# Global effects of land-use intensity on local pollinator biodiversity

Joseph Millard [1,2✉], Charlotte L. Outhwaite [1], Robyn Kinnersley[1], Robin Freeman[2], Richard D. Gregory [1,3], Opeyemi Adedoja [4], Sabrina Gavini [5], Esther Kioko [6], Michael Kuhlmann[7,8], Jeff Ollerton [9], Zong-Xin Ren [10] & Tim Newbold [1]

Pollinating species are in decline globally, with land use an important driver. However, most of the evidence on which these claims are made is patchy, based on studies with low taxonomic and geographic representativeness. Here, we model the effect of land-use type and intensity on global pollinator biodiversity, using a local-scale database covering 303 studies, 12,170 sites, and 4502 pollinating species. Relative to a primary vegetation baseline, we show that low levels of intensity can have beneficial effects on pollinator bio-diversity. Within most anthropogenic land-use types however, increasing intensity is asso-ciated with significant reductions, particularly in urban (43% richness and 62% abundance reduction compared to the least intensive urban sites), and pasture (75% abundance reduction) areas. We further show that on cropland, the strongly negative response to intensity is restricted to tropical areas, and that the direction and magnitude of response differs among taxonomic groups. Our findings confirm widespread effects of land-use intensity on pollinators, most significantly in the tropics, where land use is predicted to change rapidly.

[1] Department of Genetics, Evolution & Environment, University College London, London, United Kingdom. [2] Institute of Zoology, Zoological Society of London, London, United Kingdom. [3] RSPB Centre for Conservation Science, RSPB, The Lodge, Sandy, United Kingdom. [4] Department of Conservation and Marine Sciences, Cape Peninsula University of Technology, Cape Town, South Africa. [5] INIBIOMA, CONICET-Universidad Nacional del Comahue, Rio Negro, Argentina. [6] Zoology Department, National Museums of Kenya (NMK), Nairobi, Kenya. [7] Zoological Museum, Kiel University, Kiel, Germany. [8] Department of Life Sciences, Natural History Museum, London, United Kingdom. [9] Faculty of Arts, Science and Technology, University of Northampton, Northampton, United Kingdom. [10] Key Laboratory for Plant Diversity and Biogeography of East Asia, Kunming Institute of Botany, Chinese Academy of Sciences, Kunming, PR China. ✉email: joseph.millard.17@ucl.ac.uk

Pollinating species, particularly insect pollinators, are reported to be in decline, with change in land-cover, land-use intensity, and climate thought to be the primary drivers[1–7]. In the media, insect pollinator biodiversity change has been reported to constitute an ecological Armageddon[8]. However, the evidence on which these claims are made is patchy and contested, based on studies with low taxonomic and geographic representativeness[9–13]. For pollinators more broadly, declines have been reported in wild bees, honeybees, hoverflies, butterflies and moths, flower-visiting wasps, birds, and mammals (see Ollerton[14] for a summary of the evidence), but comprehensive studies of change tend to be biased towards North America and Europe[15], which are unlikely to be globally representative[16]. Moreover, even within well-studied taxonomic groups and regions, the magnitude and direction of change can vary depending on methodological approach, spatial scale, and metric of biodiversity change[17,18]. Recent research indicates there may be pollinator information for other geographic regions and taxonomic groups, previously untapped in synthetic analyses of pollinator biodiversity change[19]. Given the value of animal pollination to the global economy, at an estimated \$235–577 billion US dollars per annum[6], further research considering multiple metrics of biodiversity, across a broader spectrum of taxa and geographies, is required.

The reliance of global crop production on animal pollinators makes pollinator biodiversity research highly relevant to policymakers. More than 75% of globally important food crops are at least partially reliant on animal pollination, including fruits, vegetables, coffee, cocoa and almonds[20]. Three recent policy initiatives demonstrate recognition from the international community that pollinator biodiversity change represents a significant problem, and needs to be addressed: 1) the EU Pollinators Initiative called for improved knowledge of declines and causes, action to tackle drivers, and raised awareness across society on the importance of pollinators[21]; 2) the International Pollinator Initiative plan of action aims to coordinate global action for pollinator conservation[22]; and 3) more broadly, the draft post-2020 framework of the Convention on Biological Diversity describes the need for the sustainable use of biodiversity to support the productivity of ecosystems[23].

Much of the Earth's terrestrial surface is subject to anthropogenic use. More than 75% of the terrestrial world exhibits direct evidence of historical or current transformation[24], with just over 50% (~67 million km²) currently used by humans[25]. This area is comprised of ~44% for agriculture and forestry, and ~7% for infrastructure including urban areas[25]. Within both natural and disturbed land-use types, intensity of human use varies markedly. Broadly capturing the inputs used in managing land, high-intensity farming refers to a suite of technological practices designed—although not always successfully—to increase yield[26]. Treatments of the land are often in the form of chemical applicants, such as pesticides, fungicides, herbicides, and fertilisers, as well as mechanical management (tillage). Such intensive agricultural practices are commonplace in much of the modern world[27,28].

Anthropogenic land-use and land-use intensity are interrelated drivers of pollinator biodiversity change[6,29,30]. Much of the research investigating land-use effects on pollinator biodiversity has demonstrated the importance of landscape-level habitat composition, often as distance to natural habitat[31] and distance to managed land[30,32], or habitat fragmentation[33] and edge density[10]. Land-use intensity effects are generally typified by chemical application[3,34–37]. Pesticides such as neonicotinoids have been a focal point of study, given their association with declining bee populations[3,34,35] honey bee health[36], and bumblebee behaviour[38]. Other chemical inputs such as fungicides and herbicides have also been subject to investigation, tending to have indirect effects on pollinator biodiversity by increasing pesticide toxicity[37,39] and reducing floral diversity[40]. Similar indirect effects have also been shown for fertiliser application. For example, nitrogen-based fertilisers reduce plant species diversity[41], and dispense with the requirement for clover field crop rotation, further reducing floral availability for pollinators[42].

Pollinator response to landscape-level land use is mixed, with the magnitude and direction of change differing among taxonomic groups. For example, some bees, butterflies, syrphid flies, and nectarivorous pollinating birds have been found to favour open, intermediate-level forested areas of semi-natural grassland or agroforestry[43–46]. Similarly, species rich and abundant wild bee communities have been found in urban environments, indicating that for some species anthropogenic activity can be beneficial[47]. For both open and urban areas, benefits to pollinators can in part be attributed to floral availability[42,47]. More broadly however, differences in pollinator response are often attributed to traits[48–51], such as dietary specialism, mobility, and nesting behaviour[52,53]. Trait data are not available for many pollinating species, but given that phylogeny to some extent predicts traits, one would expect broad differences in response among taxonomic groups[54].

Differences in pollinator response to intensity are also likely between tropical and non-tropical regions. There are a number of reasons why this is the case. First, temperate non-tropical regions have a longer history of agricultural activity, which may have acted to filter more sensitive species[55], meaning more recent shifts towards intensive agriculture may have a smaller effect. Second, with the exception of high latitude Arctic pollinators[56], tropical biodiversity has been reported to be more sensitive to the effects of climate change[57], which may magnify the effect of land-use[58]. In terms of insect pollinators specifically, tropical insects are thought to exist closer to their thermal tolerance limits, meaning small magnitude changes in temperature have a disproportionate effect on biodiversity[59]. Third, functional specialisation tends to be higher in tropical pollination systems (i.e. there is a narrower breadth of visitors to a flower across broad taxonomic levels), which may also relate to community sensitivity to land-use change. Although recent research has addressed patterns of overall biodiversity change between geographical zones[60,61], for pollinating taxa the extent to which response to land-use intensity differs between tropical and non-tropical regions is unclear.

Here we present a global space-for-time synthesis of pollinator responses to land-use intensity. We test for global differences in responses among land-use types, taxonomic groups, geographic regions, and biodiversity metrics. We do so using two global compilations of data: 1) The PREDICTS database, a global compilation of site-level ecological survey data across different land uses and land-use intensities[62]; and 2) a new database of animal species judged to be pollinators (see Millard et al.[19] and Methods). Our final dataset includes 3862 invertebrate and 640 vertebrate species identified as potential pollinators, across 303 studies and 12,170 sites, primarily across North and South America, Europe, and Africa. We hypothesise that land-use intensity decreases site-level biodiversity (species richness, Simpson diversity, and total abundance) for pollinating species overall, but that response differs between taxonomic orders, and is more negative in the tropical zone than elsewhere. Specifically, we answer three questions related to land-use intensity and global pollinator biodiversity: 1) What are the overall effects of land-use intensity on pollinator biodiversity across all land-use types? Then focusing on croplands, for which there is the most extensive data: 2) How does the effect of land-use intensity on pollinator biodiversity differ between tropical and non-tropical areas? and 3) How does pollinator response to land-use intensity vary among

taxa? Within an anthropogenic land-use type, we show that intensity often decreases pollinator biodiversity. But relative to the primary vegetation minimal-use baseline, pollinator biodiversity is often greater at low and intermediate levels of intensity, suggesting that some level of disturbance can be beneficial. Across taxa within croplands, we again show a mixed response to intensity, varying according to both facet of intensity and taxonomic group. Specifically in the tropics however, pollinators appear highly sensitive to land-use intensity; a situation that may worsen as intensive agriculture continues to expand.

## Results

**Pollinator dataset**. We identified 1013 possible pollinating genera across 3974 abstracts in the initial automatic search of the pollination literature. After reading the abstracts associated with each genus, we confirmed 545 genera as likely pollinators at confidence levels 1–4. These 545 genera represented 141 unique families, of which 46 families, 10 subfamilies, and 5 tribes were judged to consist entirely of pollinators. Whilst consulting literature prioritised by the automatic search, we also identified an additional 51 genera with direct pollination evidence, which we assigned a confidence level between 1 and 4, and 18 additional families with extrapolated evidence. After building our set of potentially pollinating species, we then sought the opinion of 7 experts in pollination ecology (authors OA, SG, EK, MK, JO, Z-XR, and also Dr. Manu Saunders; see Supplementary Table 1), and removed or added taxa according to their suggestions. Filtering all expert-assessed pollinators from the PREDICTS database returned records for 4502 species in total, sampled at 12,170 sites. After selecting only sites in which land-use intensity and type were recorded in the PREDICTS database, a total 8,639 sites remained. Site coverage was highest in Europe (26.2%), North America (24.4%), and Africa (20%), and lowest in South America & the Caribbean (12.3%), Oceania (15.2%), and Asia (8.6%) (Fig. 1).

**Effect of land use and land-use intensity on global pollinator biodiversity**. Increasing land-use intensity from minimal to intense use was associated with a significant change in pollinator biodiversity (species richness, $F = 9.4384$; total abundance, $F = 4.8075$, $p < 0.01$; Simpson diversity, $F = 11.6691$, $p < 0.01$; Fig. 2; see Supplementary Table 3 for ANOVA tables of land-use intensity and type fitted separately). Land-use type was also a significant predictor (species richness, $F = 8.9440$; total abundance, $F = 8.0346$, $p < 0.01$; Simpson diversity, $F = 4.4150$, $p < 0.01$; see Supplementary Table 3), although declines occur more strongly within a land-use type as opposed to among land-use types. Relative to the primary vegetation minimal use baseline, for both natural and anthropogenic land-use types, biodiversity was often higher at low intensity (Fig. 2). Indeed, with the exception of cropland and young secondary vegetation, all land-use types had species richness and total abundance significantly greater than the baseline, for at least one of low or intermediate intensity (Fig. 2).

Effects of land-use intensity were strongest in urban areas, with a 43% reduction for species richness and 62% for total abundance, between minimal and intense use. Plantation forest also experienced strong declines, decreasing by 38% for species richness. For anthropogenic land uses the weakest effects of land-use intensity were seen in pasture and cropland. Species richness did not decline significantly for pasture—although there was a 75% decline for total abundance—or for cropland for both total abundance and species richness. Young secondary vegetation did not significantly differ for total abundance, but species richness declined between minimal and high intensity by 16%. All

other secondary-vegetation types (mature and intermediate secondary vegetation) did not show significant differences in pollinator biodiversity among intensity levels (Fig. 2). The AIC value for our Simpson diversity LUI model was greater than the intercept-only model, meaning it was excluded from further analysis (see Supplementary Table 4). A zero-inflated negative binomial model for total abundance did not markedly change our predictions (Supplementary Fig. 1). Similarly, neither did jack-knifing total abundance and species richness by continent (Supplementary Fig. 2), including environmental covariates (Supplementary Fig. 3), or controlling for abundance in our measure of species richness (Supplementary Fig. 4). There was significant spatial-autocorrelation in the residuals of only a small proportion of studies (2.33% of species richness studies and 4.65% of total abundance studies; Supplementary Fig. 5).

**Effect of land-use intensity on pollinator biodiversity within croplands**. Land-use intensity had a divergent effect on cropland pollinator biodiversity between the non-tropical and tropical geographical zones (Fig. 3). In the non-tropical zone, species richness and total abundance did not differ significantly among cropland intensity classes, and were significantly higher in minimal-intensity cropland compared to the primary-vegetation baseline. In contrast, in the tropical zone, species richness and total abundance decreased between primary vegetation and high-intensity cropland by 44 and 49%, respectively. Forest cover for the primary vegetation baseline did change the magnitude of response relative to cropland, with relatively bigger declines from a low forest cover baseline, although the relative difference within cropland remains largely unchanged (Supplementary Fig. 6). Greater variation in non-tropical areas is not predicted by high sample size (Supplementary Fig. 7), and response for the main crop pollinators is likely consistent with all pollinators (Supplementary Fig. 8). The AIC value for our Simpson diversity zone model was greater than the intercept-only model, meaning it was excluded from further analysis (see Supplementary Table 4). Response to total fertiliser application rate between the non-tropical and tropical zones was also insignificant for all of species richness, total abundance, and Simpson diversity (Supplementary Table 4), meaning it was excluded from further analysis.

Increasing land-use intensity in croplands had varying effects among taxa (Fig. 4). Relative to primary vegetation, abundance declines at high intensity for the invertebrate pollinators were greater than 70% for all orders, and as high as 80% for the Lepidoptera and Diptera. The most consistent invertebrate declines were in the Lepidoptera, exhibiting a negative response across a gradient of intensity for species richness, total abundance, and Simpson diversity. For flies on the other hand, relative to minimally used cropland, intermediate levels of intensity were associated with higher species richness and total abundance. For the vertebrates, the Apodiformes exhibited a strong negative response to land-use intensity, declining by at least 20% from medium-intensity cropland to primary vegetation for all three metrics (high-intensity cropland was not sampled for this taxonomic group). The Passeriformes experienced a significant reduction from the baseline to high intensity, of 30% for total abundance, 36% for species richness, and 26% for Simpson diversity.

Response to total fertiliser application rate in surrounding cropland landscape differed strongly in magnitude and direction (Fig. 5). Both Hymenoptera and Lepidoptera showed a strong negative response to increasing fertiliser application rate for both species richness and total abundance. In particular, an increase of 1000 kg/ha in fertiliser application rate was associated with a reduction of 44% in hymenopteran total abundance, whereas

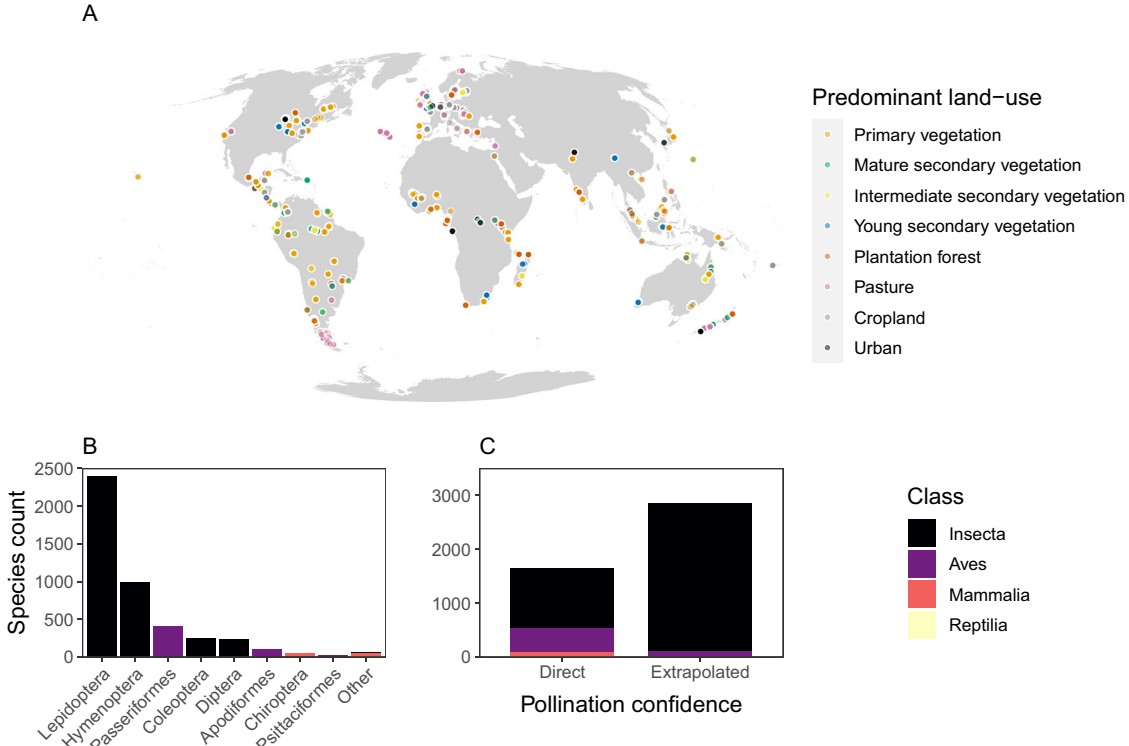

**Fig. 1 Data on species in the PREDICTS database identified as likely pollinators, after automatic text-mining, manual filtering, and expert consultation.**
**A** The global distribution of PREDICTS sites containing likely pollinating species, for which both the land-use type and intensity of that site are known ($n_{sites}$ = 8639). Colours represent land-use type: orange (primary vegetation), green (mature secondary vegetation), yellow (intermediate secondary vegetation), blue (young secondary vegetation), dark orange (plantation forest), pink (pasture), grey (cropland), and black (urban). **B** The taxonomic distribution of likely pollinating species in PREDICTS for all sites ($n_{species}$ = 4502). The number of species indicated here will be an underestimate of the number of pollinating species in PREDICTS, since this figure only includes records for which there is a full scientific binomial in the database. Some biodiversity records in the PREDICTS database are recorded above the level of species. **C** The source of information (direct evidence at the genus level or extrapolated to groups based on information for groups at higher taxonomic levels) for pollinators in PREDICTS, broken down by taxonomic class. In both **B** and **C**, there are four taxonomic classes: Insecta (black), Aves (purple), Mammalia (red), and Reptilia (yellow). The reptiles are represented by only 5 species with 'Direct' confidence (see Supplementary Table 2).

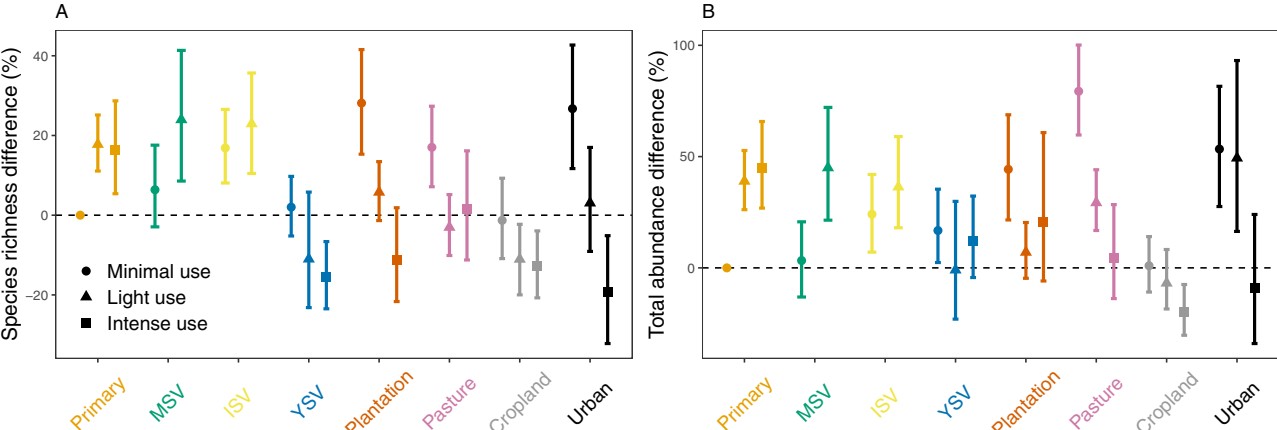

**Fig. 2 Responses of pollinator biodiversity to LUI (a combined variable of land-use type and intensity).** Each panel represents a linear or generalised linear mixed-effects model for: **A** species richness; and **B** total abundance. We excluded Simpson diversity here since AIC was greater for the main model than the intercept-only model. Colours represent land-use type: orange (primary vegetation, Primary), green (mature secondary vegetation, MSV), yellow (intermediate secondary vegetation, ISV), blue (young secondary vegetation, YSV), dark orange (plantation forest, Plantation), pink (Pasture), grey (Cropland), and black (Urban), and point shape represents land-use intensity: circle (minimal use), triangles (light use), and squares (intense use). Effect sizes were adjusted to a percentage by drawing fixed effects 1000 times based on the variance-covariance matrix, expressing each fixed effect within each random draw as a percentage of the baseline (primary vegetation minimal use), and then calculating the median value (shown as points) and 2.5th and 97.5th percentiles (shown as error bars). See Supplementary Table 5 for the number of sites and Supplementary Tables 6 and 7 for the model summaries. Source data are provided as a Source Data file.

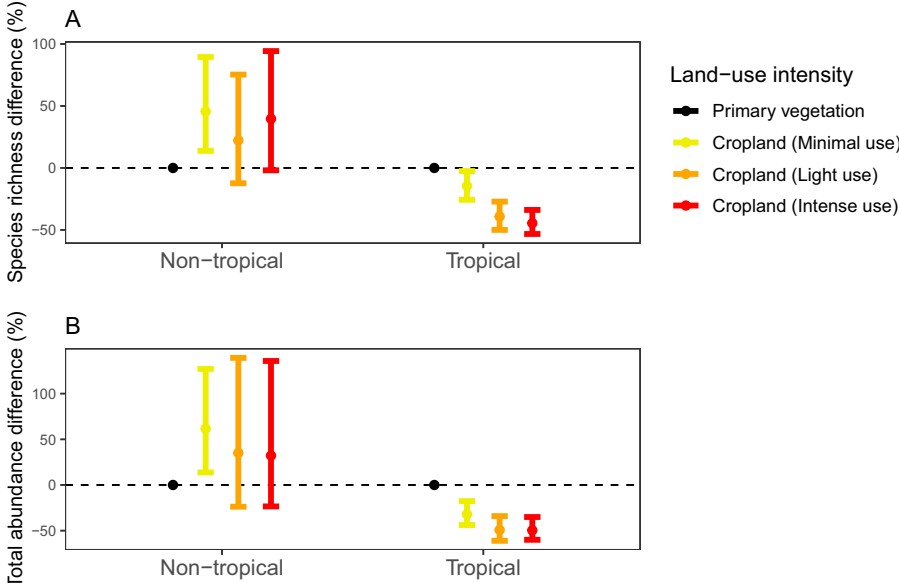

**Fig. 3 Response of pollinators to land-use intensity on cropland, for non-tropical and tropical sites.** Each panel represents a linear or generalised linear mixed-effects model for a given biodiversity metric: **A** species richness; and **B** total abundance. We excluded Simpson diversity here since AIC was greater for the main model than the intercept-only model. Colours represent the land-use intensity level, with primary vegetation (minimal use) as the reference factor: black (primary vegetation, minimal use); yellow (cropland, minimal use), orange (cropland, light use), and red (cropland, intense use). Effect sizes were adjusted to a percentage by sampling fixed effects 1000 times based on the variance-covariance matrix, expressing each fixed effect as a percentage of the value in primary vegetation for that geographical zone, and then calculating the median value (shown as points), and 2.5th and 97.5th percentiles (shown as error bars). See Supplementary Table 8 for the number of sites and Supplementary Tables 9 and 10 for the model summaries. Source data are provided as a Source Data file.

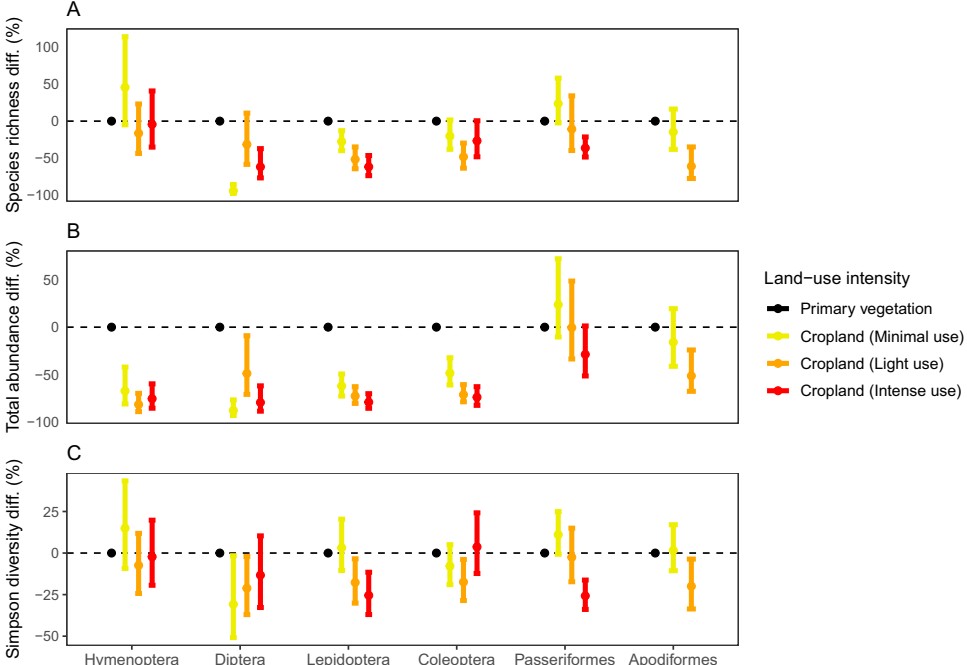

**Fig. 4 Response of different pollinator groups to land-use intensity in cropland.** Each panel represents a linear or generalised linear mixed-effects model for a given biodiversity metric: **A** species richness; **B** total abundance; and **C** Simpson diversity. Colours represent the land-use intensity level, with primary vegetation (minimal use) as a reference factor: black (primary vegetation, minimal use); yellow (cropland, minimal use), orange (cropland, light use), and red (cropland, intense use). Effect sizes were adjusted to a percentage by sampling fixed effects 1000 times based on the variance-covariance matrix, expressing each fixed effect within each random sample as a percentage of the value in primary vegetation for that taxonomic order, and then calculating the median value (shown as points), and 2.5th and 97.5th percentiles (shown as error bars). See Supplementary Table 11 for the number of sites and Supplementary Tables 12, 13, and 14 for the model summaries. Source data are provided as a Source Data file.

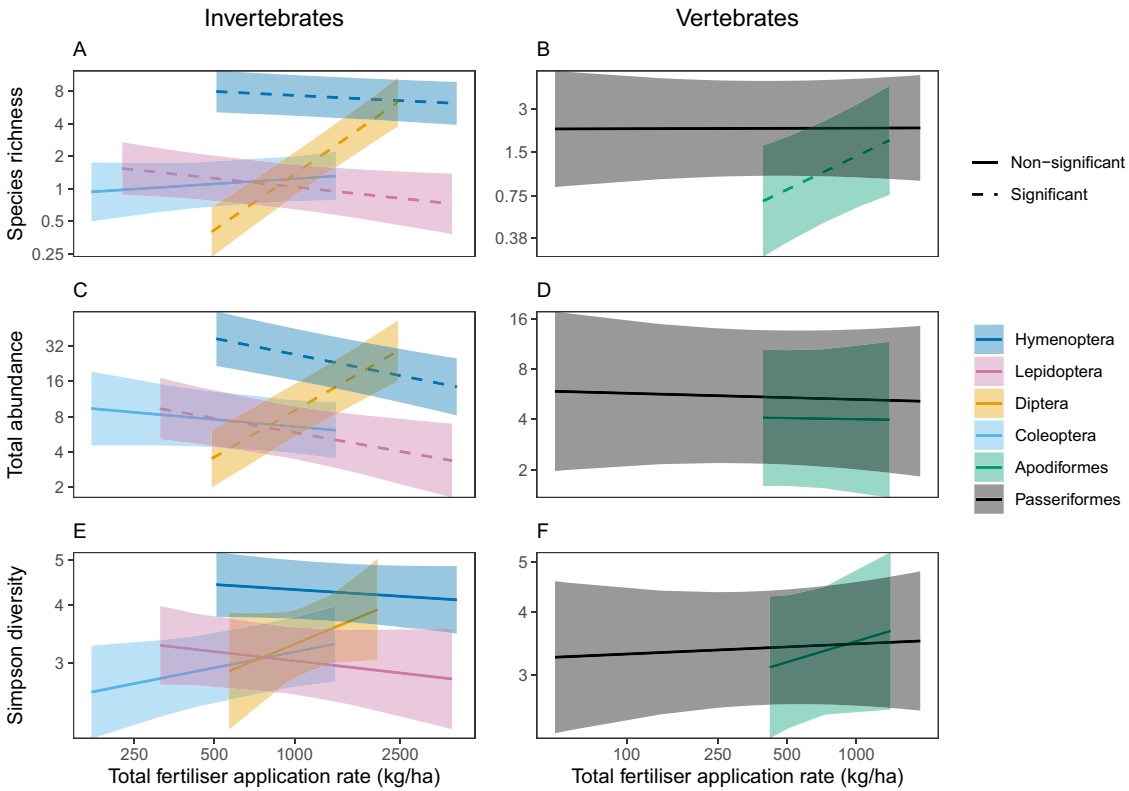

**Fig. 5 Response of pollinator biodiversity in cropland sites to total fertiliser application rate in the landscape (kg/ha) (note that each metric is plotted on a log scale), predicted across 95% of the range of sampled fertiliser values for each taxonomic order.** Each panel represents a linear or generalised linear mixed-effects model for a given biodiversity metric for four invertebrate orders: **A**, **C**, and **E** invertebrate species richness, total abundance, and Simpson diversity respectively; and **B**, **D**, and **F** vertebrate species richness, total abundance, and Simpson diversity respectively. Coloured lines represent median fitted estimates for each taxonomic order, with shading representing 95% confidence intervals: light blue (Coleoptera), light orange (Diptera), dark blue (Hymenoptera), pink (Lepidoptera), green (Apodiformes), and black (Passeriformes). Dashed lines represent significant interactions ($p < 0.05$) between taxonomic order and total fertiliser application rate, and solid lines non-significant interactions. See Supplementary Table 15 for the number of sites, Supplementary Tables 16, 17, and 18 for the model summaries, and Supplementary Fig. 9 for the global distribution of total fertiliser application rate. Source data are provided as a Source Data file.

lepidopteran abundance fell 50% over the same range. Dipteran richness and abundance, on the other hand, increased markedly by 760% and 374% respectively. Coleopteran response to total fertiliser application rate was insignificant for all of species richness, total abundance, and Simpson diversity. For the vertebrates, the Apodiformes increased by 163% for species richness, whereas the Passeriformes experienced no marked change for any of species richness, total abundance, or Simpson diversity. Although the AIC value for our Simpson diversity model was less than the null model (Supplementary Table 4), all interactions between total fertiliser application rate and taxonomic order for Simpson diversity were insignificant (Fig. 5).

## Discussion

Using a space-for-time approach, here we showed that land-use intensity is associated with significant changes (both positive and negative) in pollinator species richness, total abundance, and Simpson diversity, particularly for invertebrate pollinators. This study represents, as far as we know, the largest global analysis of the response of animal pollinator biodiversity to land-use type and intensity, and the first to consider large-scale differences in responses among taxa, geographic regions, and biodiversity metrics. Our results are consistent with previous analyses showing reductions in overall pollinator biodiversity at high land-use intensity[63], and increases at low-intermediate intensity[64]. In particular, low-intensity urban sites (villages and green spaces[62])

have higher pollinator biodiversity than the primary vegetation baseline, but at high intensity urban pollinator species richness is significantly lower than the baseline (although this was not the case for abundance). This is concordant with previous research demonstrating that urban areas can support species-rich and abundant pollinator populations[47,65]. We also highlight that strong negative responses to land-use intensity within croplands are largely restricted to the tropics, with no apparent effect (and even increases at low intensity) in non-tropical cropland. This is an important result, given the dominance of animal pollinated plants in tropical environments[66], and that rapid agricultural expansion is predicted to occur in the near future[67]. Furthermore, we show pronounced differences in response among taxonomic groups, consistent with time-series studies showing differential trends among UK invertebrate taxa[68]. Pollinator biodiversity change resulting from land-use intensity may have consequences for pollination[69] and crop yields[70], especially in the tropics: Although the abundance of some pollinating groups (i.e. flies) is greater on intensively fertilised cropland, increases may not compensate for overall losses across other pollinating groups.

Differences in response to intensity between tropical and non-tropical areas are likely driven by the interacting effects of historical land-use and climate sensitivity, which differ between the temperate and tropical zones. Non-tropical temperate regions have a long history of anthropogenic land-use, which has likely filtered out many sensitive species[55], meaning that contemporary differences in land use and land-use intensity may be weakly

associated with pollinator biodiversity. Indeed, historical land-use has been shown to be strongly associated with current species richness and abundance of insect pollinators[54], and may contribute towards an overall increase in pollinator biodiversity at low land-use intensity[71]. The tropical zone, on the other hand, has a shorter history of intensive agricultural land-use[72], meaning recent intensification has stronger effects on pollinator community composition. Tropical biodiversity is also thought to be more sensitive to the effects of climate change[59], which may be exacerbating the effects of land use[58]. Further research is required to tease out the relative contribution of historical land-use and climate change to tropical pollinator biodiversity.

Pollinating insects across multiple geographic locations and taxonomic groups have been reported to have declined, particularly for biomass and abundance[13,73]. In our space-for-time analysis, we found significantly lower abundance on high-intensity cropland—relative to primary vegetation—for all insect orders, especially for the hymenopterans, lepidopterans, and coleopterans. We also found significant reductions in insect abundance and richness in response to increasing fertiliser application rate on croplands, particularly for the hymenopterans and lepidopterans. Sensitivity to land-use for butterflies has previously been attributed to dietary specialism: relative to flies, many butterflies are known to be dietary specialists as larvae[4], making a reduction in lepidopteran species richness and diversity likely when plant species richness is reduced, which is known to occur at high fertiliser application rates[3,41]. Previous research in the temperate zone has indicated beetle flower visitors are sensitive to land-use change[74]. Although we found a decrease in coleopteran abundance relative to the primary vegetation baseline, response to fertiliser application rate was mixed and insignificant. It may be that fertiliser application buffers against the more negative effects of associated intensity, since some pollinating coleopterans are known to have a larval preference for fertile soils[75]. For the hymenopterans, sensitivity to land-use has been reported previously, particularly for solitary bees[69] which are on average highly specialised[74].

Dipteran abundance and species richness increased with fertiliser application rate, concordant with previous studies demonstrating increased dipteran biodiversity on managed land[7,74,76]. Dipteran resilience to land use and land-use intensity has been attributed to a number of traits, including low dietary specialisation on floral resources[74], high mobility[77], absence of parental care[76], and larval preference for agricultural habitats[76]. Syrphid fly larval development in agricultural land is particularly of note. Semi-aquatic syrphid larvae are known to favour eutrophic or manure-contaminated habitats[78], which is consistent with the strong positive gradient for fly richness and abundance in response to increasing fertiliser application rate. In contrast, the response of fly abundance and species richness to our overall measure of land-use intensity was negative, suggesting that fertiliser application does not sufficiently capture all aspects of land-use intensity on cropland. Neonicotinoids such as imidacloprid, for example, have adverse effects on flies, given their association with declines in insects in general[34], and visitation rate in flies specifically[79]. As an additional analysis, we used the PEST-CHEMGRIDS global estimation of pesticide application rate[80] to build additional models of pollinator response between tropical and non-tropical croplands. We found that the response for flies to pesticide application rate is in fact similar to fertiliser (Supplementary Figs. 10 and 11), the cause of which is unclear, although it may be that fertiliser and pesticide application rate are correlated. Further analysis is required to tease out the relative effects of fertiliser and pesticide application at the global scale.

Compared to invertebrates, vertebrate pollinators appear to be less sensitive to the effects of land-use intensity, particularly with respect to change in total abundance. Relatively higher resilience to land-use intensity has been found for vertebrate pollinators in tropical forests[46], and suggested in broad comparisons across taxa[4]. However, although previous work found that vertebrate resilience diminishes after controlling for study design[4], our results suggest that vertebrate pollinators are indeed less sensitive than invertebrate pollinators to increasing land-use intensity. Such relative vertebrate resilience likely relates to body size and mobility[81], both of which are typically greater in vertebrate pollinators.

Change in global pollinator biodiversity resulting from land-use intensity may have consequences for crop pollination. If the loss in pollination service provided by sensitive crop-pollinating taxa (Hymenoptera) cannot be offset by gains in more resilient taxa (Passeriformes), then the service will experience a net loss. Four lines of evidence indicate losses in sensitive taxa may not be buffered by more resilient taxa: first, response to land-use intensity of the main crop-pollinating groups (bees, flies, beetles, wasps, thrips, birds, and bats) appears consistent with responses across all pollinators (Supplementary Fig. 8); second, there is evidence from multiple historical localities that significant pollinator deficits can result from the losses associated with intensive agriculture[82,83], irrespective of differences among taxonomic groups; third, crop-yield reduction has been linked to changes in pollinator biodiversity[32,84], and fourth, relative to bees, those species which are increasing (i.e. dipterans) are known to contribute less to crop pollination[85].

Our analysis of pollinator biodiversity change is subject to limitations. First, the nature of our study as a space-for-time analysis means we may overlook extinction-debt effects. Such effects can be controlled for by assessing change over time at a specific location or region[13,73]. However, at the global scale, given that long-term studies are lacking, space-for-time analyses represent a necessary alternative[86]. Second, most of the results we present here are relative to a baseline of primary vegetation with minimal human use, which inevitably varies in nature, especially between tropical and non-tropical sites (Supplementary Figs. 12 and 13). In particular, we might expect non-tropical primary habitats to be more open than tropical primary habitats, which is likely to be more favourable to pollinators, and thus might partly explain the absence of responses to land-use intensity in non-tropical areas. Indeed, we show that the margin of reduction from a low forest-cover baseline is greater than from a high forest-cover baseline (Supplementary Fig. 6). Nevertheless, the overall responses for tropical and non-tropical areas remain unchanged. Third, we recognise that the explanatory power of our models (see Supplementary Table 19 for pseudo R-squared values) is low, with the random effects explaining a large degree of variation. However, we emphasise that the aim of our analysis was not to predict pollinator biodiversity for a specific location, but rather to investigate general trends in the direction and magnitude of change. Fourth, we analyse raw species richness which we recognise may be confounded by abundance. However, since our validation model predicting estimated species richness (using the Chao1 estimator) does not differ markedly from raw richness, a confounding effect of abundance is unlikely (Supplementary Fig. 4). Fifth, our dataset of pollinator biodiversity responses is spatially biased towards the non-tropics, particularly Europe and North America. Given that tropical pollinators are affected more negatively, our overall results therefore likely underestimate the impact of land-use intensity. However, our continental jack-knife for our overall LUI models showed that the exclusion of any of the Americas, Europe, or Africa (the continents for which we have the most sites) did not markedly influence our predictions (Supplementary Fig. 2). Moreover, our additional validation analysis in which we re-sampled 1000 sites from each of the

tropics and non-tropics would indicate that greater variation in non-tropical regions is likely not predicted by greater sample size (Supplementary Fig. 7). Sixth, total fertiliser application rate was estimated at a relatively coarse spatial scale, which for some pollinators—particularly those such as insect pollinators which respond strongly to more localised change—will not be meaningful. Since spatial scale is known to predict pollinator response to land-use intensity[87], we could infer a different response if more localised fertiliser estimates were available globally. Seventh, we recognise that evidence for a given species will not always be representative of all species in a whole genus, or all life-history stages within a species. For example, the species *Crocidura cyanea* has been found to feed on and carry pollen, but most other Crocidurans are insectivorous[88]. As a result, pollination confidence for many species in the genus *Crocidura* will be less than the genus-level evidence would imply. In the main, we assumed evidence for a single species would be representative of whole genera, given the association between phylogeny and traits[54]. We also reasoned that for some insect groups, searching at the species level would be ineffective, given the large number of species with little evidence. Eighth, species confirmed as pollinating one flowering plant will not necessarily make an important contribution to all flowering plants, or to the pollination of crops.

Anthropogenic activity has significantly altered the biosphere. Such changes have had, and will continue to have, profound consequences for animal pollinator biodiversity. Here we show significant pollinator biodiversity change in response to land-use intensity, with both negative and positive effects. It is likely that climate change will drive further changes in pollinator biodiversity, particularly for insects which respond strongly to changes in ambient temperature[5]. Further research is required to better resolve the way in which these threats interact at the global level. For crop pollination services in the tropics, the repercussions of land-use and climate change could be great, with a growing body of evidence indicating high wild pollinator biodiversity is required to sustain productive yields[84]. Although the complexities of this relationship are not yet fully understood, there is sufficient evidence to suggest that pollination shortfalls in the tropics could result from continuing anthropogenic intensification and expansion.

## Methods

**Pollinator dataset construction.** We built an animal pollinator dataset through a semi-automatic approach, combining an automatic text-analysis method (see Millard et al.[19]) with manual inspection of the automated output. Here we describe the full methodology used to derive this dataset, and in the supplementary information release the final set of taxa both sampled in PREDICTS and identified as potential pollinators (Supplementary Data 1).

We first created a list of possible pollinating animal genera through automatic text analysis of the pollination literature. We used an initial automated search to avoid biasing towards well-known pollinators, and to markedly reduce the input required in searching. We considered the pollination literature to be any primary research article published in English returned through a search for the term 'pollinat*' in Scopus, and which mentioned an animal species in the abstract. We considered a possible pollinating genus to be any animal genus appearing as part of a Latin binomial in a pollination-related abstract returned from Scopus. Genus scraping was accomplished using the Taxonfinder and Neti Neti algorithms implemented in the taxize R package, with animal species confirmed through a series of character string matches to the Catalogue of Life (see Millard et al.[19] for a detailed methodology).

For each possible pollinating genus, we then read the abstracts in which these animals appeared, searching for evidence confirming that genus as pollinating. For any situation in which the abstract was inconclusive, we also searched the full text of the paper. For each confirmed pollinating genus, we then assigned a level of confidence between 1 and 4 based on the type of evidence, following Ollerton and Leide[89]: 1) experimental evidence confirming pollination; 2) evidence of pollen carrying; 3) evidence of nectar/pollen feeding; 4) evidence of non-destructive/non-predatory flower visitation. We read abstracts for each genus searching for the highest level of evidence, either until we could be sure that the confidence value should be 1, or we ran out of abstracts for that genus. Non-destructive flower visitor refers to any animal which visits a flower without causing damage to the plant. This meant the exclusion of ants, which are typically referred to as poor pollen vectors, given that they damage pollen through secretions from the meta-pleural gland[90]. Non-predatory flower visitor refers to any animal which visits for some purpose other than predation. This meant the exclusion of animals such as crab spiders, which predate on pollinators during visitation, and therefore contribute minimally or negatively to pollination[91]. We did not classify broad statements as evidence for pollination—for example, one study stated that *Phylidonyris novaehollandiae* is a key pollinator[92]—unless it was associated with specific evidence reinforcing that statement, or some claim that pollination in that genus is well-known or widely acknowledged.

Given that we only had direct evidence for a sample of all pollinating genera, we then searched for higher-level groups of likely pollinators. From the confirmed pollinators in the original list of genera, we identified all unique families with at least one pollinator. For each family, we assessed the breadth of evidence for pollination through consulting the abstracts and taxonomic group reference books (see Supplementary Data 2). For any family with evidence of pollination across multiple branches of that family, and no evidence of any species definitely not pollinating, we assumed that the whole family is pollinating. If unable to extrapolate across the whole family, we then searched progressively lower taxonomic groups (i.e. subfamily, tribe, subtribe), searching for the point at which we could be relatively confident that the entire group contributes to pollination. If unable to extrapolate for a given group, we kept only the genera with direct evidence. For example, within the family Macroscelididae (elephant shrews), we found only one genus (*Elephantulus*) with pollination evidence, and no evidence across the rest of the family, meaning we kept only that genus.

To compile the final list of pollinators, we merged all genera identified directly as pollinators, and then all taxonomic groups identified indirectly, with all biodiversity records in the PREDICTS database. Any record in the PREDICTS database not for a pollinating genus or extrapolated taxonomic group was thereby filtered out. As a result, specific sites or studies were only kept if they were represented by at least 1 pollinator record. Merging direct evidence pollinators first means any species is picked up at its highest level of confidence, and only assigned one confidence value. PREDICTS does not record additional taxonomic ranks between family and genus, so for any species extrapolated at a taxonomic level below the level of family (i.e. subfamily, tribe, subtribe), we consulted compiled genera lists for each group, using taxonomic references (see Supplementary Data 2) and Wikispecies, and then filtered these genera from PREDICTS. As an additional check of our final list of likely pollinating species, we sought the opinion of experts in pollination ecology. We sent lists of the likely pollinating genera in PREDICTS to 7 experts (OA, SG, EK, MK, JO, Z-XR, MS; see Supplementary Table 1), and removed any taxa identified as highly unlikely pollinators (see Supplementary Data 1 for the final list of likely pollinating genera in PREDICTS).

**Effect of land use and land-use intensity on global pollinator biodiversity.** We used the PREDICTS database to model responses of animal pollinators to land-use type and intensity[62]. The PREDICTS database is structured such that each site is nested at a series of levels (Supplementary Fig. 14), allowing one to account for variation owing to study methodology. The database contains variables for land-use intensity (minimal, low, and high) and land-use type (primary vegetation, mature secondary vegetation, intermediate secondary vegetation, young secondary vegetation, plantation, pasture, cropland, and urban). Land-use intensity for each land-use type is defined according to a series of variables, such as fertiliser and pesticide application, mechanisation, and hunting (see Newbold et al.[93] for more details).

After merging the PREDICTS database with our set of likely pollinating species we performed a series of data-processing steps. We removed any sites for which land-use type and land-use intensity was unknown. We also removed sites in secondary vegetation at an unknown stage of recovery. We combined the factors for land-use intensity and type to create a single variable (henceforth referred to as LUI), following the methodology of De Palma et al.[16]. After combining land-use intensity and type, we then removed the class "Mature secondary vegetation-Intense use", which was represented by only 5 sites, and "Intermediate secondary vegetation-Intense use", which was represented by only 23 sites. After removing these factors, site representation was ≥43 sites for all land use type and land-use intensity combinations (see Supplementary Table 5). We then calculated site-level species richness (the number of uniquely named species sampled at a site), Chao1-estimated species richness (the number of species at a site controlled for abundance[94]), total abundance (the sum of all species sampled abundances at a site), and the Simpson diversity index (the reciprocal of the sum of squared proportional abundances for all species sampled at a site). Sampling effort was accounted for by dividing the abundance values for each measurement by the sampling effort (rescaled to a max value of 1 for each study) for that record, as in De Palma et al.[16]. For any subsequent analyses we worked only from the sampling effort adjusted measurements. Given sampling effort adjustments, and that raw abundances were in some cases measured as densities, many total abundances will be non-integer values.

We built generalised linear mixed-effects models with a Poisson error distribution for species richness and Chao1-estimated species richness[94], and linear mixed-effects models for Simpson diversity and total abundance. In an initial set of models all biodiversity metrics were fitted as a function of land-use intensity, land-use type, and their interaction, for all likely pollinators in PREDICTS (see

Supplementary Table 3). We then built a set of models predicting each biodiversity metric as a function of LUI, for the same set of pollinators. We did not use a generalised model with Poisson errors for total abundance or Simpson diversity because most recorded measurements are not integer counts of individuals. Instead we $\log_e$-transformed all total abundance and Simpson diversity values (adding one because of zero values) to normalise the model residuals. Due to the nested nature of the database (see Supplementary Fig. 14 and Hudson et al.[62]), we included a random intercept of study identity to account for variation in sampling methods, sampling effort and broad geographical differences among studies, and a random intercept of spatial block within study to account for the spatial structuring of sites. An additional (observation-level) random intercept of site identity was included in the species richness model, to control for the over-dispersion present in species richness estimates[95]. Random-effects structures were selected to minimise AIC values. We checked for overdispersion in the species richness models using the function GLMEROverdispersion in the R package StatisticalModels. We compared each model against an intercept-only model, and discarded any main model for which AIC was greater than the null model (see Supplementary Table 19 for pseudo R squared values for all significant models).

We carried out a series of additional validation analyses for our set of LUI models. 1) We checked for study-level spatial autocorrelation in the residuals of any significant model, using the Moran's I test (Supplementary Fig. 5). 2) We checked the extent to which a negative binomial zero-inflated model for total abundance would have differed from a linear model approach (Supplementary Fig. 1). 3) We checked the extent to which the fixed effects would have differed if we had fit a model with climatic variables as potentially confounding covariates (Supplementary Fig. 3), including both the maximum temperature of the hottest month and the total precipitation of the wettest month—both over the 12 months previous to the end data of each sample—which have previously been indicated as important biological variables[58]. 4) We jack-knifed the sites for each significant model by continental region to check the extent to which geographic biases influenced our predictions (Supplementary Fig. 2). 5) We checked the extent to which an abundance-controlled estimate of species richness (Chao1-estimated species richness[94] would have differed from species richness alone (Supplementary Fig. 4).

**Effect of land-use intensity on cropland pollinator biodiversity**. We focused on cropland in our remaining analyses, given the importance of animal pollination to crop production. We built 3 models for all potential pollinating species, modelling each of three biodiversity metrics (species richness, total abundance, and Simpson diversity) in cropland as a function of land-use intensity (minimal, low, high), geographical zone (temperate/tropical), and their interaction. We included minimally used primary vegetation in these models as a baseline. Given that the structure of this baseline differs among sites—particularly between tropical and non-tropical areas (see Supplementary Figs. 12 and 13)—and that this may affect our predictions for pollinator biodiversity, we also built a set of models with a high (≥60% cover) and low (≤40% cover) forest cover baseline, using Hansen et al.[96] forest cover data (Supplementary Fig. 6). We also carried out two additional validation analyses. 1) We checked whether unequal site number between our tropical and non-tropical data predicted the size of our 95% confidence intervals. Specifically, we resampled 1000 sites from each of the tropical and non-tropical sites a total of 100 times, and then for each group of 2000 (tropical and non-tropical) fitted total abundance as a function of land-use intensity, geographical zone, and their interaction. We then plotted the distribution of the size of the 95% confidence intervals for all models (Supplementary Fig. 7). 2) We checked whether response to land-use intensity between the tropics and non-tropics would have been the same if we had analysed only the main crop-pollinating groups (Supplementary Fig. 8; i.e. bees, wasps, beetles, thrips, flies, birds, and bats; see OIlerton[85]).

We also built 3 models for a vertebrate and invertebrate cropland subset of the database, modelling the same biodiversity metrics as a function of land-use intensity, taxonomic order, and their interaction, again including minimally used primary vegetation as a baseline. Our taxonomic subset included the better-sampled invertebrate orders Hymenoptera, Lepidoptera, Diptera, and Coleoptera, and the vertebrate orders Apodiformes and Passeriformes, and represented 3006 sites in total (see Supplementary Table 11). For both our geographical zone and taxonomic order models, we selected from the same set of random-effects structures (as in the main models), aiming to minimise AIC values. We tested each model against an intercept-only model and a model with one fixed effect for land-use intensity, and discarded any main model for which AIC was greater than the null model (see Supplementary Table 4).

We also explored the effect of a continuous variable describing a specific aspect of land-use intensity (fertiliser application rate) on pollinator biodiversity, specifically for cropland. We used Earthstat fertiliser data[97,98] at a spatial scale of 5 × 5 min—equivalent to 10 × 10 km at the equator—largely for the years 1999–2000, aggregated as the total application in kg per hectare for nitrogen, phosphorous, and potassium on 17 major crops (see Supplementary Table 20 for the full list crops). We aggregated the Earthstat fertiliser data by summing the per hectare application rate rasters for all crop/fertiliser combinations, and then extracting the summed fertiliser values at each site (see Supplementary Fig. 9, note that site-level geographic distribution is lesser relative to our overall pollinator

biodiversity models). Given that the spatial scale of this aggregated application rate data is greater than that of specific sites, our fertiliser metric refers to application rate in the surrounding landscape, rather than at that specific site. We chose to use fertiliser data given its availability at the global scale, reasoning that its application would both drive change itself, and broadly act as a surrogate for intensity. We built models for all potential pollinating species, modelling each of three biodiversity metrics (species richness, total abundance, and Simpson diversity) in cropland as a function of log10(fertiliser application rate +1), geographical zone (temperate/tropical), and their interaction. We also built models for the invertebrate and vertebrate subset (Hymenoptera, Lepidoptera, Diptera, Coleoptera, Apodiformes, and Passeriformes), modelling each of three biodiversity metrics as a function of total fertiliser application rate, taxonomic order, and their interaction. We compared each model against an intercept model and a model with one fixed effect for total fertiliser application rate, and excluded any main model for which AIC was greater than the null model. As a supplement to our total fertiliser application rate analyses, we also used PEST-CHEMGRIDS[80] to build an analogous set of total pesticide application rate models (Supplementary Fig. 11). PEST-CHEMGRIDS represents 20 of the most common pesticides for 6 individual crops and 4 aggregated crop groups, again at a scale of 5 × 5 min (Supplementary Fig. 10). All analysis and data processing were carried out in R v.4.0.3. Source data for all model predicted values are included as a Source Data file.

**Reporting summary**. Further information on research design is available in the Nature Research Reporting Summary linked to this article.

## Data availability
All data sets required to replicate this study are available online. We used the PREDICTS database for biodiversity records of pollinating species and site-level categorical factors of land-use type and intensity (https://doi.org/10.5519/0066354). Our subset of pollinating species in the PREDICTS database is available on FigShare (https://doi.org/10.6084/m9.figshare.12815669.v2). We used EarthStat fertiliser application rate data to calculate site-level fertiliser application for the years 1999–2000 (https://doi.org/10.1038/nature11420 and https://doi.org/10.1126/science.1246067). We used WorldClim 2.1 to calculate potentially confounding climatic variables (https://www.worldclim.org/data/index.html). We used Hansen et al[96] data to calculate site-level forest cover (https://doi.org/10.1126/science.1244693). We used PEST-CHEMGRIDS to calculate site-level pesticide application rate (https://doi.org/10.1038/s41597-019-0169-4). Source data for all figures are provided with this paper. Source data are provided with this paper.

## Code availability
All data compilation, cleaning, and analysis were carried out in R v4.0.3. All code used is available on Zenodo via GitHub (https://doi.org/10.5281/zenodo.4593493)[99].

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

## Acknowledgements

J.M. was funded by the London NERC DTP—award number NE/R012148/1—and the RSPB on a CASE studentship. This work was supported by a grant from the UK Natural Environment Research Council (grant ref. NE/R010811/1) and by a Royal Society University Research Fellowship awarded to T.N. Thanks to Manu Saunders for comments on an earlier draft of the manuscript, and for checking the inclusion of our invertebrate pollinators. Thanks to Monica Ortiz for aggregating the forest cover data in the supplementary information. Thanks to Bruna Lacerda da Silva Abreu for comments on an earlier draft of the manuscript.

## Author contributions

J.M. and T.N. conceived and designed the study. J.M. led the analysis and produced the figures with C.O. and R.K. J.M. wrote the manuscript with input from T. N., R.G. and R.F. O.A., S.G., E.K., M.K., J.O. and Z-X.R. assessed the list of pollinating species and edited the manuscript. All authors approved the final version of the manuscript.

## Competing interests

The authors declare no competing interests.
