## [Peer Review File · Nature Communications]

REVIEWER COMMENTS

Reviewer #1 (Remarks to the Author):

The study addresses a key question of our times: the decline of pollinators. The main argument for doing this new analysis is that the existing information is patchy taxonomically and geographically. One of the main findings is that land use intensity has a reverse relation with pollinator richness. I think the argument is true, just as for all other taxa and ecosystem functions, so rather general. The finding is well established already, most evidence and synthesis (e.g. IPBES Pollinator Assessment, Ollerton 2017 Annual reviews – both are cited) showed it. I have to note that this well-established finding actually not fully supported by the data, as for flies it simply the opposite, that is not all the four invertebrate taxa showed the reverse relationship between land use intensity and diversity indices.

In summary, the paper has an important topic, strengthens the known evidence, may interests the wider scientific community. The interpretation not always fits the results. What I am missing are further, novel issues like geographical variations (beyond the divide of temperate versus tropical), and these variations across taxa.

Introduction.

All what is written is true. Decline of pollination and the patchy evidence. Reading it, however, I do not feel that there is a need for a new analysis, as no new primary data are presented. Well known open database was used, available for several years already, even before the IPBES assessment. I simply do not expect to decrease patchiness.

Methods

Search for pollinators. This is an interesting "reverse" search for pollinators. Naively I would start to search the life history of species, species descriptions and so on, as such literature are more basic on who pollinate. This reverse way can be indirect, for example, most papers addressing pollinators and land use intensity will list some bee species, but not because they know that bee species are pollinate, but because they read it in the life history literature. Thus, to find pollinator animal name in a paper is not necessary evidence that the species pollinate, unless the study do scrutinize pollination itself. I do not say this method is wrong, just that the collected pollinator names have to be checked.

Results and discussion

Fig 6 shows decline for two taxa and increase for other two in invertebrates. Still the general conclusion is that increasing land use intensity is associated with diversity decline – thus it is not fully supported by the results.

The comparison between tropical and temperate zones has important message. I would go further with the analysis (instead of repetitively show that LUI can destroy pollinator communities), and do a more thorough analysis of possible geographical differences.

Line 42-43: IPBES pollination report has a similar, but more up-to-date estimation from 2017.

L.48-57. Three policy initiatives are listed, two from the EU, so it has some conflict with the aim of the paper to provide geographically better balanced evidences. There are many more policy initiative, Promote Pollinators, for example.

L112. typo, two dots.

L116-121. Confusing to say four questions, then numbering 1. 2. and 3.

L 306. 51 genera were not found by automatic search, which is c. 10% plus. Considerable amount - I wonder if the 545 genera from the automatic search really needed an automatic search? I have some uncertainty regarding the method, see above.

L387 and below. The relationship of pollinators and fertiliser application has much restricted geographical coverage (Fig 2 in SUP) compared to the pollinator sites, thus have limits for generalisation.

L392 and elsewhere: thousands kg/ha fertiliser application is much higher value than usual in very intensive croplands. Seems to be wrong (e.g.

<http://www.fao.org/tempref/docrep/fao/009/a0787e/A0787E00.pdf>).

L454-56. Fig 6 shows decline for two taxa and increase for other two in invertebrates. This simply

not support this statement.

L 468-71. Coleoptera is missing, where species richness and diversity increased as well (Fig 6). So can not state only Diptera is increasing.

L481. delete REF

L484-85. et al. is italics or not? use the same style in the whole manuscript.

L551-52. In deed. Regarding geographical representation no development compared to previous studies.

Reviewer #2 (Remarks to the Author):

This is a comprehensive study which I read with great interest. The authors report negative impacts of land-use intensity on pollinator abundance and richness. Such a negative impact on biodiversity in general and on pollinators in particular is not new, but the authors confirm this by providing evidence at a global scale, for multiple pollinator taxa and land use types. With this breadth the study is highly relevant for global pollinator conservation. The study is based on the best global dataset available, but I have some concerns which should be addressed.

(i) At several places, the impression is generated that the authors analyse temporal changes, especially when they first or subsequently refer to papers on temporal trends when referring to their own results (e.g. L 491 ff). Although the authors explicitly mention that they use a space-for-time approach, this needs to be made more clear and such an impression should be avoided.

(ii) The authors entirely focus on land-use intensity, but especially at a large geographic scale, causality might be questioned. Land-use intensity can be confounded by other factors such as climate or topographic heterogeneity. The authors include study ID as a random effect which might only partly control for such effects, since the single studies might encompass larger gradients in climate or heterogeneity themselves. The authors should check and, when necessary, control for potentially confounding effects.

(iii) The authors aim at drawing global conclusions but, as they mention, the geographic distribution of the data likely leads to a bias towards North America and Europe. Instead of only raising a note of caution, they should rely on approaches to avoid this bias, such as (repeated randomized) spatial thinning.

(iv) The authors use three biodiversity metrics: abundance, richness and Simpson diversity, but do not make use of this when interpreting their results. Usually, abundance and richness are highly correlated and it would make sense to correct richness for abundance, e.g. with asymptotic richness estimates. With this, a more mechanistic understanding of the impacts of land-use intensity on pollinators would be possible, i.e. does Land-use intensity basically affect abundance which is only reflected by declines in richness, or is there also a direct effect on richness (and abundance distribution)? However, it seems that the actual richness per data point is on average very small (about 3?) which might impede diversity estimates. In such a case the authors might proof the possibility to aggregate the data to higher levels of their nested design. It is also interesting that the authors did not find any effects in Simpson diversity, which, together with abundance and richness, should be interpreted in terms of abundant and rare species.

(v) The authors make the statement that the impact of land-use intensity is more prominent than that of land-use type. But this is based on visual interpretation of the figures but not formally tested. For this they should not combine land-use type and land-use intensity into one factor but keep them separated in the models and compare their importance. This would also allow for testing interaction effects between both, helping to identify in which land-use type the impact of intensification is strongest. Further, land-use intensity is categorical with levels from low to intermediate to high, which makes it a good candidate for an ordered variable. The authors should consider this, or at least explain why the disregard it.

(vi) The authors need to be more precise in the description of their results (see detailed comments).

(vii) The discussion would need some structural improvement, since it is not always clear to which of their results their conclusions refer (all pollinators in different land-use types, comparison with baseline vs. along intensity, only cropland). The authors also need to be much more differentiated in the discussion and should avoid any overselling. E.g. in L 454 or L 472 they state that they found an overall negative response to land-use intensity, while this was only the case for five out of eight land-use types (Fig. 1), for the tropics and not for temperate regions (Fig. 2), and for birds only in one case in agricultural landscapes (Fig. 3), and was evident for abundance and richness but not for Simpson diversity.

(viii) It would also be worth to discuss their results not only in terms of increasing land-use intensity but also in comparison to the baseline (primary vegetation with minimal use) which is the intuitive interpretation of the figures. The authors should be clear that minimal (and light) use, i.e. extensive use, can indeed have positive effects on pollinator diversity (many valuable habitats are purely anthropogenic but with extensive use, at least in Europe and the US). Actually, only intense use in young secondary vegetation, cropland and urban sites led to a lower diversity compared to primary vegetation (Fig. 2). This would enrich the discussion, not only focusing on the devastating effects of land-use but also highlighting the potential beneficial effects of extensive use.

(ix) Birds are entirely missing from the discussion. Their (non-)responses should be compared to invertebrates.

(x) References to the supplement need to be updated. Some are likely wrong and others are simply missing.

Detailed comments.

L15 Only values for urban are provided in the text (unless I missed them). Where do the others come from? Can you provide a table in the supplement?

L18 One example for potentially misleading the reader. Your findings do not confirm studies on temporal losses of pollinators, since you did not analyse temporal trends. Your found relationships might be one explanation for observed losses.

L89 This is not true. Extensive anthropogenic land use can increase pollinator diversity (e.g. as we transformed forests into highly heterogeneous pastoral landscapes very early in Europe).

L137 Please provide the number of papers.

L143 Please explain COL

L149 Check brackets

L169 Please mention the names of the experts.

L207 In the previous paragraphs you describe very detailed the assessment of the uncertainty in the assignment of being a pollinator, but for me it is unclear how this confidence was used for the generation of the final species list.

L225 The merging process is not clear. In particular, did you keep sites even when the studies did not focus on pollinators? This would lead to an excess of unreliable zeros. Or did you define 'pollinator studies' and disregard all the others?

L232 Please provide total number of sites.

L233 Table S2?

L238 You control for sampling effort for abundance. If possible do the same for richness and Simpson diversity (see earlier comment).

L242 It is only important that the residuals are normally distributed.

L242 Log-transformation is suboptimal (O'Hara et al. 2010). Please use a GLMM with proper error distribution (e.g. negative binomial or gamma). For abundance you can directly control for sample effort in such a binomial model.

L243 About the zeros, see my earlier question and if many, you might consider a zero-inflation model.

L249 Your random effect controls for the nested structure but not necessarily for the spatial structure. Therefore, you should check for spatial autocorrelation and control for it if necessary.

L252 All models? This is not relevant for linear models.

L260 This justification is rather weak. If this is the case, why did you do it (Fig. 2)? I acknowledge that conducting detailed analyses per taxonomic groups for all land-use types would heavily increase the results and the space for discussion needed. However, we already know quite a lot about cropland, but much less on other land-use types. It would be really interesting to expand this analysis at least to pastures and urban areas (or any other subset for which you have a good reason).

L272 Table S3?

L281 Table S20. Many supplementary table are not referred to at all.

L298 Which statistical software did you use?

L299 Please provide measures of goodness of fit for your models in the results.

L336 Please differentiate better for richness and abundance.

L337 and 339 Please check the percentages carefully. It seems that you are drastically overestimating the reduction. It seems that you calculated the percentage as the respective range in Fig. 2. E.g. for abundance change in urban sites: roughly 54% (MU) + 14% = 68%. If this is the case, the difference is reported relative to the baseline instead of the highest value (or lowest intensity) of the respective land use type.

E.g. baseline = 10, Urban MU = 15.4 (+54% to baseline), Urban IU = 8.6 (-14% to baseline), span in percent is 68% as reported for abundance, but Urban IU is only a 44% reduction relative to Urban MU (difference = 15.4 - 8.6 = 6.8, which mean a 44% reduction from 15.4).

L372 Without error bars for primary vegetation, diversity is higher in minimal use. Please also refer to the respective supplementary tables here and for all subsequent results.

L375 To avoid the impression that you had significant effects please change to: "weak and insignificant". Since there was no effect, there is no need to show this figure.

L377 Here you are mixing comparisons to baseline and among increasing land-use intensity. Please be more precise. Compared to baseline, you found negative effects in most of the cases, but there is also a positive effect (Hymenoptera, richness). For increasing land-use intensity you found a negative effect only for Hymenoptera for abundance, and a negative effect for Lepidoptera and Passeriformes and a positive for Diptera for richness.

L380 All three biodiversity metrics? You said that there was no effect for Simpson diversity.

L384 Please stick to the error bars, there was no effect.

L387 Why do the biodiversity metrics for the same group not cover the same range in fertilization in Fig. 6?

L387 This was not the case. The relationship was positive for most taxa and only negative for Hymenoptera and Lepidoptera.

L391 ff Please provide % changes as you did before and not x-fold changes. Also, make the fertilizer ranges comparable, i.e. refer to changes per 1000 kg/ha for instance.

L396 Results are missing for the fact that Coleoptera, Apodiformes and Passeriformes increased in most metrics.

L455 Simpson was not significant in most of the cases (only for fertilizer in croplands).

L461 The response in the tropics was not much greater than in temperate regions (suggesting that there was an effect in the temperate zone), but it was restricted to the tropics.

L463 Do you mean wild plant pollination? There are other areas which are pollination-dependent for crops.

L471 You may refer to some homogenization literature.

L472 You did not find an overall negative response to land-use intensity (see earlier comments). You need to be much more differentiated, and especially clear when you refer to comparisons with the baseline and when to differences between different levels of intensity independent of the baseline.

L476 Diversity was not maintained it was considerably increased.

L478 This is a big jump and some restructuring might help.

L480 It would be worth to mention that during the long history of land-use in temperate zones most anthropogenic activities led to an initial increase in biodiversity by generating new habitat types and increasing landscape heterogeneity. Such positive effects are also visible in your results, e.g. when minimal or light use lead to an increase in diversity. You should not only focus on the negative effects but also discuss the positive effects.

L481 Delete "REF"

L486 Do you have a reference for the shorter history?

L489 2002 instead of 202

L491 Do not mix support for temporal declines with your space-for-time results. "we found strong declines [...] on high intensity cropland" and referring first to temporal declines generates the impression that you confirm temporal declines, which is not the case.

L494 Not clear what you are referring to, baseline or increasing intensity? Actually, you found declines only for Hymenoptera and an increase for Diptera. Compared to baseline, Diptera declined the same as the others.

L505 But why do they respond positively with fertilization? Could this be related to larval diet?

L508 Please be more precise. There was no effect for richness (instead of 'less severe') and even a positive for minimal use compared to the baseline.

L515 Two out of four is not 'most'. In addition to Diptera, Coleoptera increased as well (richness and Simpson, likely no response in abundance).

Fig. 1 Please provide information about land-use intensity by colouring the dots e.g. according to LUI or if too messy only to land-use intensity.

Fig. 2, 3 and 5 Please provide percentiles also for the baseline.

Table S7 P-values are missing

O'Hara RB, Kotze DJ (2010) Do not log-transform count data. *Methods in Ecology and Evolution* 1: 118-122.

Reviewer #3 (Remarks to the Author):

In their study "Global effects of land-use intensity on local pollinator biodiversity", the authors gathered data from 340 studies covering >5000 species judged to be pollinators to model the effects of land-use type and land-use intensification on pollinator species richness, abundance and diversity. Using this very impressive dataset, the authors find that while land-use type only weakly affected pollinator biodiversity, increasing land-use intensity negatively affected pollinator richness and abundance across all land-use types. Moreover, declines in intensified cropland were found to be greater in tropical than in temperate regions, with specialized pollinator clades such as bees and butterflies being particularly negatively affected.

While the individual findings are not totally surprising or outstandingly novel, this study definitely is one of the (if not the) most comprehensive analysis and attempts at drawing general conclusions on the effects of human land-use on pollinator biodiversity at a global scale. Thus, this manuscript may represent a very important contribution to the literature that should attract a great number of readers and interest from ecologists, conservationists and members of the general public interested in pollinator conservation – a very important topic given the great reliance of wild plants and global crop production on animal mediated pollination.

My main concern with this study is the authors' usage of a baseline to which to compare pollinator responses in richness/abundance/diversity with changing land-use type and land-use intensity (see Figures 2+3). In the respective models, the authors used pollinator data from study sites classified as "minimally used primary vegetation" as a baseline. The results are then presented as %-differences to this baseline (e.g., 20% lower or higher species richness in a given land-use type as compared to primary vegetation). At first glance, this appears to be a very intuitive way to present the data, and to compare pollinator responses across different biodiversity metrics (richness, abundance, Simpson diversity) and across land-use types of varying intensity levels. However, I wondered if primary vegetation is in all cases the ideal baseline for such a %-level comparison. What is striking is, for example, that, on average (i.e. across intensity levels) many land-use types have much higher richness and/or abundance than sites covered with primary vegetation (see e.g. Figure 2). Negative effects of human land-use mainly become apparent at higher levels of land-use intensity. Figure 3 suggests that such positive effects were mainly driven by datasets from the temperate region, where (minimal to light) land-use often supports higher pollinator biodiversity than primary vegetation. It remains unclear how primary vegetation was defined in these temperate studies, but I suspect these are mainly studies conducted in forested ecosystems (e.g., pollinator studies conducted in central and northern Europe). However, given that much of Europe is dominated by cultural landscapes characterized by open ecosystems, I wonder whether choosing such forested ecosystems as the reference for the pollinator biodiversity baseline is really the best choice. For example, the vast majority of endangered bees in central Europe are adapted to open habitats (that are of anthropogenic origin and have been cultivated as such often for centuries), such as dry grasslands, orchards or areas with high amount of bare ground for nesting (often on agricultural land). By comparison, pollinator biodiversity is usually low in forested ecosystems (which is sometimes even the case in the tropics, where flower visitor activity is often much higher at forest edges and in forest clearings because of a lack of flowering undergrowth vegetation in closed rainforests). Hence, while I understand the appeal of having a baseline measurement for pollinator biodiversity, I would like to see more justification for the choice of (and the habitat types hidden behind) primary vegetation, especially in the context of temperate ecosystems with a long history of human land-use influence.

In addition, as most of the results are currently expressed as %-differences (to a given baseline). This obscures a bit the underlying absolute differences, which would be nice to see. For example, in Figure 4, model predictions for absolute measures of pollinator richness, abundance and diversity are presented. I was surprised to see that these predictions are, however, on a relatively low level (e.g., averages of species richness of pollinator in temperate cropland ranging between ca. 3.5 and 2.5 species, and in the temperate zone between 3.5 and 4 species; Fig. 4A). This made me wonder how large the absolute differences in species richness and abundance actually are (e.g., in Figs. 2+3+5), when not expressed on a %-differences scale.

Other comments:

L17: In the discussion flies are presented as a pollinator group that is relatively robust to land-use, so I was surprised to see them mentioned here together with the more sensitive bees, butterflies, etc.

L29: What do you mean with "mainstream media"? I wonder about the choice of wording (but I am not a native speaker – am just curious because of Trump's usage of such words).

L42: Perhaps include "US" in addition to "dollars" to clarify.

L47: Not all of the 75% of globally important food crops do fully rely on animal pollination (some show only very modest yield increase), but they do so at least partially (and some fully). I suggest including "at least partially" before "rely" to make this clear.

L89-96: In addition, pollinator increases in open habitats may simply be driven by greater availability of floral resources and nesting sites (for ground-nesting pollinators, e.g., 66%-75% of solitary bees in central Europe).

L101-103: In addition, there remain uncertainties associated with taxonomic and geographic biases that further hinder general conclusions on pollinator responses to land-use change globally (de Palma et al. 2016, Scientific Reports)

L112: Please also include the numbers of studies included from these geographic regions. Looking at Figure 1, you did a remarkable job to also include as many studies from Africa, Asia and South America as possible. Nonetheless, there still appears to be a geographic bias in favor of the global North (as is usually unavoidable in such global datasets).

L115: I agree with the hypothesis that pollinator biodiversity may respond more negatively to land-use (intensification) in the tropics, but the point was not very well developed in the introduction. You briefly mention the potential greater susceptibility in the tropics (L98-101). In addition, you may also wish to include arguments from your discussion, e.g., the point that temperate ecosystems have been shaped by humans for a much longer time and in a more persistent fashion than most tropical regions. Think of cultural landscapes (small fields, orchards, meadows, hedgerows) that harbor much greater biodiversity than modern agricultural landscapes (simplified and intensively used cropland) or natural ecosystems (forests), and which in fact even favored a great number of synanthropic species. By contrast, tropical biodiversity is often not adapted to human land-use.

L237: Was this index of sampling effort used in any of the following models? I could find mentioning of random effects, but not of this variable.

L245: It surprised me that most recorded measurements of pollinator abundance were not integer values – where these then expressed as densities, or why was this?

L254-255 + L273-275 + L294-296: I could not make sense of these sentences "We tested each model against an intercept model, and excluded any biodiversity metric for which deltaic was greater than ~2". With "biodiversity metric", do you mean pollinator

richness/abundance/diversity? And how do you exclude these? (you can exclude predictors from your model but not the response variable – i.e., the biodiversity metric?). I believe you mean you discarded the model in case the null model (including only the intercept) performed better (based on the AIC score)?

L261: I do not agree that fertilizer application is less important in urban environments. Particularly in the tropics, urban farming plays a big role for food security – and here, of course, also fertilizers are applied, if available. In general, I was surprised that not all land-use intensity levels were compared across all land-use types – although lack of data may certainly be an explanation.

L265-266: Please see my thoughts concerning this choice of baseline above.

L276: I very much liked the additional focus on fertilizer rate effects in cropland. This made me wonder whether other such data are available, e.g., on pesticide application (at a global scale)? This would be particularly interesting, as these additional analyses provide a much more direct link of land-use intensification to pollinator biodiversity – which is a bit harder to grasp when only presented with the land-use intensity classes.

L303ff: Honestly, I found the whole system of classifying the pollinators into confidence levels rather confusing. This was certainly necessary and important when gathering the study dataset. But the classification is hard to grasp – because higher numbers do not necessarily indicate higher certainties (the opposite appears to be rather the case), and because there appears to be a break in the confidence changes between 1-4 and 5.1-5.4 (so no real logical progression from 4 to 5.1). Lastly, I wondered if you also tried some sensitivity tests of your model outcomes based on different levels of pollinator confidence – e.g., only using data with pollinator classifications based on high levels of confidence?

L311-312: Please also provide numbers or %-values of sites across these geographic regions, so that differences in geographic coverage become clearer.

L332: Figure 1 - Reptiles are impossible to see in panel C – perhaps use a log-scale instead.

L425: Figure 2 – Percentiles are much greater for predictions in the temperate sites. Is this only due to larger variation because of the greater number of study sites (as compared to number of tropical study sites), or also a result of more divergent pollinator responses in temperate regions?

L440: Figure 4 – Are all of these model predictions statistically significant? If not, please use dashed lines to indicate non-significant relationships.

L452: Figure 6 – Again, I wondered about significant vs. non-significant effects.

L461: Word too much/missing in this sentence.

L462: Greater reliance of what? Angiosperm plants in general, crops, or both?

L468: Perhaps worthwhile to nonetheless mention studies that highlight the importance of flies and other non-bee pollinators for crop pollination (e.g., Rader et al., PNAS).

L481: Delete “REF”.

L489: Full publication year for Williams et al. missing.

L512: This is an interesting finding, and good that you did this additional analysis. Could you also quantify the “much stronger response for abundance” here in the main text – e.g., by expressing the difference in %?

L546ff: I appreciate the discussion of study limitations. As outlined before, it would be important to also briefly discuss the choice of the baseline habitat, and how this may affect biodiversity comparisons to anthropogenic land-use types. Even though of anthropogenic origin, many

agricultural habitat types with minimal usage are of highest conservation concern (e.g., dry grasslands), and harbor many more rare and common pollinator species (e.g., wild bees) than natural ecosystems with primary vegetation.

Response to reviewers

Thanks very much all three reviewers for your constructive, useful, and positive comments on our manuscript. Your comments have helped to improve the robustness of our analysis, and to acknowledge better the nuances of pollinator biodiversity change among taxa, geographic regions, biodiversity metrics, land-use types, and levels of intensity. See below for our response to each of your points in turn (in red). Please also note that all line numbers refer to changes in the manuscript with tracked changes toggled off.

Reviewer 1

The study addresses a key question of our times: the decline of pollinators. The main argument for doing this new analysis is that the existing information is patchy taxonomically and geographically. One of the main findings is that land use intensity has a reverse relation with pollinator richness. I think the argument is true, just as for all other taxa and ecosystem functions, so rather general. The finding is well established already, most evidence and synthesis (e.g. IPBES Pollinator Assessment, Ollerton 2017 Annual reviews – both are cited) showed it. I have to note that this well-established finding actually not fully supported by the data, as for flies it simply the opposite, that is not all the four invertebrate taxa showed the reverse relationship between land use intensity and diversity indices.

In summary, the paper has an important topic, strengthens the known evidence, may interests the wider scientific community. The interpretation not always fits the results. What I am missing are further, novel issues like geographical variations (beyond the divide of temperate versus tropical), and these variations across taxa.

Thank very much to reviewer 1 for highlighting a series of points on the interpretation and framing of our results. Addressing these points has helped to make our study more consistent across manuscript sections.

In our revised version we have now amended the description of our results to reflect better our models (L350-443). In the Introduction and Discussion we highlight that the key novelty of our study is in investigating taxonomic and geographic differences at the global scale for multiple metrics of biodiversity (L120; L477-480). We also now explore geographical variation in more detail, including an additional jack-knife analysis showing that our results are robust to the exclusion of individual continents (Figure S3).

Introduction.

All what is written is true. Decline of pollination and the patchy evidence. Reading it, however, I do not feel that there is a need for a new analysis, as no new primary data are presented. Well known open database was used, available for several years already, even before the IPBES assessment. I simply do not expect to decrease patchiness.

Thanks to reviewer 1 for highlighting that the novelty of our study was unclear. While it is true that the PREDICTS database we use existed prior to this study, it has not been used previously to quantify the response of pollinating species to land-use intensity. Moreover, as far as we know, there are no studies of this scale comparing pollinator responses across taxonomic groups and between tropical and non-tropical regions. We now state the above in the manuscript i.e. “ we present the most comprehensive global space-for-time synthesis of pollinator responses to land-use intensity” (L120).

Methods

Search for pollinators. This is an interesting "reverse" search for pollinators. Naively I would start to search the life history of species, species descriptions and so on, as such literature are more basic on who pollinate. This reverse way can be indirect, for example, most papers addressing pollinators and land use intensity will list some bee species, but not because they know that bee species are pollinate, but because they read it in the life history literature. Thus, to find pollinator animal name in a paper is not necessary evidence that the species pollinate, unless the study do scrutinize pollination itself. I do not say this method is wrong, just that the collected pollinator names have to be checked.

Thanks for your comments on our pollinator search approach. We opted to use an initial automated search to avoid biasing towards well-known pollinators, and to markedly reduce the input required for the initial search. We now describe this in the main text (L146-147). As a check-and-balance on our pollinator search approach, we checked our list of pollinators with an international group of 7 pollination ecologists (Opeyemi Adedaja, Sabrina Gavini, Esther Kioko, Michael Kuhlmann, Jeff Ollerton, Zong-Xin Ren, and Manu Saunders), and removed any taxa that these experts deemed highly unlikely to be pollinators (L199-203). This consultation process resulted in the removal of 57 genera and the addition of 2 genera. Please see supplementary data 1 for the final list of pollinating taxa, including those either added or removed following our expert consultation (also see L144 and L203 where this data is referenced in the main text).

Results and discussion

Fig 6 shows decline for two taxa and increase for other two in invertebrates. Still the general conclusion is that increasing land use intensity is associated with diversity decline – thus it is not fully supported by the results.

We have now tempered the language in our manuscript to emphasise the heterogeneity of pollinator response between taxa and tropical/non-tropical regions (L20, L22, L476, L488-490, L493-495).

The comparison between tropical and temperate zones has important message. I would go further with the analysis (instead of repetitively show that LUI can destroy pollinator communities), and do a more thorough analysis of possible geographical differences.

Thanks for your comments on the difference in response between tropical and temperate regions. We would agree that this result is interesting. Partly in response to reviewer 3's comments on baselines, we now include an additional figure in the supplementary information investigating the effect of forest cover on pollinator biodiversity between tropical and non-tropical regions (Figure S12). We also now include an additional analysis in which we jack-knife Figure 2 for continental regions (Figure S3), showing that our overall predictions are largely robust to the exclusion of any individual continent. Given the size of the datasets, we were not able to build fine-scale divisions of response to intensity for specific geographic areas. Previous PREDICTS studies have found that the greatest part of the geographical variation in land-use responses is captured by a tropical/temperate distinction (see Newbold et al 2020, now cited on L118 in the main text).

Newbold, T., Oppenheimer, P., Etard, A. and Williams, J.J., (2020). Tropical and Mediterranean biodiversity is disproportionately sensitive to land-use and climate change. *Nature Ecology & Evolution*, 4(12), pp.1630-1638.

Line 42-43: IPBES pollination report has a similar, but more up-to-date estimation from 2017.

We have amended this reference and value to that of IPBES 2017 (L45-46).

L48-57. Three policy initiatives are listed, two from the EU, so it has some conflict with the aim of the paper to provide geographically better balanced evidences. There are many more policy initiative, Promote Pollinators, for example.

Thanks to reviewer 1 for pointing out this bias. We have now changed the second of these policy initiatives to the “The International Pollinator Initiative” to be consistent with our statement concerning efforts from the international community (L55).

L112. typo, two dots.

We have removed the extra full stop (L128).

L116-121. Confusing to say four questions, then numbering 1. 2. and 3.

We have amended this statement to say that we set out to answer three questions (L132).

L 306. 51 genera were not found by automatic search, which is c. 10% plus. Considerable amount - I wonder if the 545 genera from the automatic search really needed an automatic search? I have some uncertainty regarding the method, see above.

Thanks for your comments on our partially automated approach. We hadn't made the rationale for this approach sufficiently clear. We used an initial automated search not to find all pollinating taxa, but to avoid biasing towards well-known pollinators, and to reduce the input required in the initial search of the literature. We now describe this in the main text (L146). This initial automated approach resulted in the inclusion of at least one oft-overlooked taxon (Phelsuma), which would ordinarily be excluded in studies of pollinator biodiversity. After prioritising taxa with an automated approach, we then manually checked all taxa (see 'Methods' L156-175) Given the novel nature of our approach, we have now also consulted the input of 7 pollination ecologists to check our list of pollinators—please also see previous comment.

L387 and below. The relationship of pollinators and fertiliser application has much restricted geographical coverage (Fig 2 in SUP) compared to the pollinator sites, thus have limits for generalisation.

Thanks for pointing out this omission. In the interest of transparency, we now include references to table for the number of sites in each model (Tables S1-S5). For fertiliser application rate on cropland, we include a figure for the distribution of these sites (Figure S7). We also include an explicit statement in the methods acknowledging that the geographic distribution of our fertiliser models is less than our overall models of pollinator biodiversity (L307).

L392 and elsewhere: thousands kg/ha fertiliser application is much higher value than usual in very intensive croplands. Seems to be wrong (e.g.

<http://www.fao.org/tempref/docrep/fao/009/a0787e/A0787E00.pdf>).

Thanks for pointing out that the scaling of our fertiliser application rate data is unclear. The fertiliser data we use is the summed application rate of nitrogen, phosphorous, and, potassium for 17 crops, taken from EarthStat (Mueller et al, 2012; see L301). These values are therefore for total fertiliser

application rate at a specific location (across all crops grown), whereas the above FAO report describes average application rates for a given crop or country. Please see “Fertilizer use per hectare of cropland, 2002 to 2015” here (<https://ourworldindata.org/fertilizers>), in which the FAO reports a total fertiliser application rate of over ~30,000 kg/ha for Singapore, indicating that values in the thousands are not unreasonable.

L454-56. Fig 6 shows decline for two taxa and increase for other two in invertebrates. This simply not support this statement.

Thanks for pointing out this discrepancy. We agree that our statement here was not consistent with our results. We have amended the text to emphasise that we mean “land-use intensity is associated with significant changes, both positive and negative, in pollinator species richness, total abundance, and Simpson diversity” (L477-477). We also highlight differential trends in relation to Outhwaite et al (2012) (L493-495), and at the end of this paragraph in relation to increases for some pollinating taxa (i.e. flies) (L498-500).

L 468-71. Coleoptera is missing, where species richness and diversity increased as well (Fig 6). So can not state only Diptera is increasing.

We have removed this statement from the end of this paragraph in response to comments from reviewer 2. Coleopteran response to total fertiliser application rate was insignificant for all of species richness, total abundance, and Simpson diversity (L437). In the Results we also now emphasise varying effects of land-use intensity and total fertiliser application rate on pollinator biodiversity (L419; L431-432), which we reemphasise in the Discussion (L476).

L481. delete REF

We have deleted ‘REF’ from this citation (L505).

L484-85. et al. is italics or not? use the same style in the whole manuscript.

We have changed et al to non italics in all instances.

L551-52. Indeed. Regarding geographical representation no development compared to previous studies.

We now include a statement in the results describing that half of our sites come from outside North America and Europe (L343). We would also emphasise that—as far as we know—there are no studies analysing broad taxonomic and geographic difference for a database of this size. We now state this explicitly in the Introduction (L120) and the Discussion (L478).

Reviewer 2

This is a comprehensive study which I read with great interest. The authors report negative impacts of land-use intensity on pollinator abundance and richness. Such a negative impact on biodiversity in general and on pollinators in particular is not new, but the authors confirm this by providing evidence at a global scale, for multiple pollinator taxa and land use types. With this breadth the study is highly relevant for global pollinator conservation. The study is based on the best global dataset available, but I have some concerns which should be addressed.

Thanks very much to reviewer 2 for raising a series of important methodological points, which have helped to increase the robustness of our study.

For each of your core points (ii-v) we include an additional analysis in the supplementary information. We also mention each of these analyses in the methods, with a reference to the figures in the supplementary information.

i) At several places, the impression is generated that the authors analyse temporal changes, especially when they first or subsequently refer to papers on temporal trends when referring to their own results (e.g. L 491 ff). Although the authors explicitly mention that they use a space-for-time approach, this needs to be made more clear and such an impression should be avoided.

Thanks to reviewer 2 for pointing out that our use of a space-for-time analysis is unclear. We now include an explicit statement that our study uses a space-for-time approach in the Introduction (L120) and Discussion (L475, also see L519, L584, L587).

(ii) The authors entirely focus on land-use intensity, but especially at a large geographic scale, causality might be questioned. Land-use intensity can be confounded by other factors such as climate or topographic heterogeneity. The authors include study ID as a random effect which might only partly control for such effects, since the single studies might encompass larger gradients in climate or heterogeneity themselves. The authors should check and, when necessary, control for potentially confounding effects.

We now include an additional analysis in the supplementary information, in which we fit 2 climate variables (max temperature of the hottest month and max precipitation of the wettest month) as covariates in the models of Figure 2. Including these covariates has little effect on the land-use intensity fixed effect (Figure S5), indicating that land-use intensity is likely not being confounded by climate.

(iii) The authors aim at drawing global conclusions but, as they mention, the geographic distribution of the data likely leads to a bias towards North America and Europe. Instead of only raising a note of caution, they should rely on approaches to avoid this bias, such as (repeated randomized) spatial thinning.

We agree with reviewer 2 that our analysis would benefit from an explicit check for spatial biases. We now include an additional analysis in the supplementary information (Figure S3) in which we jack-knife Figure 2 for the continental regions Africa, Asia, the Americas, Europe, and Oceania (i.e. we remove with replacement the sites for each continent, and then rerun our models for each subset). We reasoned that spatial thinning in this manner would most explicitly check for the marked effect of single continents (i.e. the Americas or Europe). Here we show that our overall predictions are largely robust to the exclusion of any individual continent.

(iv) The authors use three biodiversity metrics: abundance, richness and Simpson diversity, but do not make use of this when interpreting their results. Usually, abundance and richness are highly correlated and it would make sense to correct richness for abundance, e.g. with asymptotic richness estimates. With this, a more mechanistic understanding of the impacts of land-use intensity on pollinators would be possible, i.e. does Land-use intensity basically affect abundance which is only reflected by declines in richness, or is there also a direct effect on richness (and abundance distribution)? However, it seems that the actual richness per data point is on average very small (about 3?) which might impede diversity estimates. In such a case the authors might proof the possibility to aggregate the data to higher levels of their nested design. It is also interesting that the

authors did not find any effects in Simpson diversity, which, together with abundance and richness, should be interpreted in terms of abundant and rare species.

Thanks to reviewer 2 for pointing out the importance of asymptotic species richness in controlling for the potentially confounding effect of abundance. We now include an additional figure in the supplementary information (Figure S6) in which we predict estimated species richness (using the Chao 1 estimator) as a function of LUI, showing that these predictions do not differ markedly from species richness alone. We introduce this additional analysis in the methods (L265), and discuss it in the context of limitations in the Discussion (L598). Although Simpson diversity was insignificant for the models in Figures 2 and 3, it was significant for both Figures 4 and 5 (Table S17). We now emphasise this point in the Results (L372-373, L413-414, L441-443).

(v) The authors make the statement that the impact of land-use intensity is more prominent than that of land-use type. But this is based on visual interpretation of the figures but not formally tested. For this they should not combine land-use type and land-use intensity into one factor but keep them separated in the models and compare their importance. This would also allow for testing interaction effects between both, helping to identify in which land-use type the impact of intensification is strongest. Further, land-use intensity is categorical with levels from low to intermediate to high, which makes it a good candidate for an ordered variable. The authors should consider this, or at least explain why they disregard it.

We have now altered the language of the abstract to avoid suggesting that differences among land-use type are relatively small, since on the basis of our analyses we cannot draw this conclusion. In the methods we now describe an additional analysis in which we fit land-use type and intensity separately (L236-238), which we then lead with in the Results (L350-356). In the supplementary information we include ANOVA tables for these models, for each of species richness, abundance, and diversity (Table S20).

(vi) The authors need to be more precise in the description of their results (see detailed comments).

We would agree that the description of our results in our original submission were at times imprecise. We have now amended all the values described in the results and addressed all your specific comments below.

(vii) The discussion would need some structural improvement, since it is not always clear to which of their results their conclusions refer (all pollinators in different land-use types, comparison with baseline vs. along intensity, only cropland). The authors also need to be much more differentiated in the discussion and should avoid any overselling. E.g. in L 454 or L 472 they state that they found an overall negative response to land-use intensity, while this was only the case for five out of eight land-use types (Fig. 1), for the tropics and not for temperate regions (Fig. 2), and for birds only in one case in agricultural landscapes (Fig. 3), and was evident for abundance and richness but not for Simpson diversity.

Thanks to reviewer 2 for highlighting the need for a restructure of our discussion. We have made a number of specific changes: In the opening paragraph, instead of “overall reduction”, we instead state that land-use intensity is associated with “significant change” (L476). We further emphasise differences in response among pollinating groups (L493-495). We have amended the opening sentence of paragraph 1 of the Discussion to more specifically describe the results of our study (L475-477), as well as the close paragraph (L425-433). We have also amended the abstract to better reflect our results (L14-23), and to be more consistent with the structure of the Discussion.

(viii) It would also be worth to discuss their results not only in terms of increasing land-use intensity but also in comparison to the baseline (primary vegetation with minimal use) which is the intuitive interpretation of the figures. The authors should be clear that minimal (and light) use, i.e. extensive use, can indeed have positive effects on pollinator diversity (many valuable habitats are purely anthropogenic but with extensive use, at least in Europe and the US). Actually, only intense use in young secondary vegetation, cropland and urban sites led to a lower diversity compared to primary vegetation (Fig. 2). This would enrich the discussion, not only focusing on the devastating effects of land-use but also highlighting the potential beneficial effects of extensive use.

Thanks for emphasising that we should consider change relative to the baseline as well as within a land-use type. We now emphasise in the results that change relative to the baseline is often positive at low and intermediate land-use intensity (L356), for both natural and anthropogenic land-use types. We also now emphasise this point at the start of the Discussion (L476), in the abstract (L14), and in the context of urban areas (L483-485), which have previously been reported as pollinator rich.

(ix) Birds are entirely missing from the discussion. Their (non-)responses should be compare to invertebrates.

We now include a separate paragraph in the Discussion on vertebrate/invertebrate differences in response (L560-568). Given the length of the Discussion in our revised version we opted not to discuss the response of the birds specifically.

(x) References to the supplement needs to be updated. Some are likely wrong and other are simply missing.

We have now checked to make sure the supplementary information is correctly referenced in the main text.

Detailed comments

L15 Only values for urban are provided in the text (unless I missed them). Where do the others come from? Can you provide a table in the supplement?

In the abstract we now explicitly state which value refers to which land-use type and biodiversity metric (L17-19). We also now include an additional supplementary data set for all the values in each of Figures 1-5 (supplementary data 2), which we reference in the main text (L326).

L18 One example for potentially misleading the reader. Your findings to not confirm studies on temporal losses of pollinators, since you did not analyse temporal trends. Your found relationships might be one explanation for observed losses.

We agree that our statement here was misleading. We have amended this to the following: "Our findings confirm widespread effects of land-use intensity on pollinator biodiversity, most significantly in the tropics, where climate and land use are predicted to change rapidly in the coming decades." (L23-25).

L89 This is not true. Extensive anthropogenic land use can increase pollinator diversity (e.g. as we transformed forests into highly heterogeneous pastoral landscapes very early in Europe).

We agree that our original statement here was not correct. We have amended this sentence to state that "Pollinator response to landscape-level land use is mixed" (L90).

L137 Please provide the number of papers.

We now state the number of abstracts containing possible pollinating genera (L330).

L143 Please explain COL

We have replaced 'COL' with the 'Catalogue of Life' (L154).

L149 Check brackets

We have removed the additional brackets here. This statement now reads 'following Ollerton and Liede (1997)' (L160).

L169 Please mention the names of the experts.

We contacted 7 pollination ecologists for input on our pollinator database: Opeyemi Adedaja, Sabrina Gavini, Esther Kioko, Michael Kuhlmann, Jeff Ollerton, Zong-Xin Ren, and Manu Saunders (see supplementary information Table S19 and L201-202).

L207 In the previous paragraphs you describe very detailed the assessment of the uncertainty in the assignment of being a pollinator, but for me it is unclear how this confidence was used for the generation of the final species list.

Thanks for pointing out that our assessment of uncertainty is unclear. In our revised version we have opted to remove the confidence classification system for extrapolated pollination. As an alternate check against uncertainty, instead we have now opted to consult a group of 7 pollination ecologists, and have removed or added taxa at their suggestion (see our previous response).

L225 The merging process is not clear. In particular, did you keep sites even when the studies did not focus on pollinators? This would lead to an excess of unreliable zeros. Or did you define 'pollinator studies' and disregard all the others?

We have now clarified in the methods that we filtered out any records from the PREDICTS database that were not for pollinating taxa (L192-193). As a result we kept only sites and studies that contained records on pollinators.

L232 Please provide total number of sites.

We now describe the total number of sites in supplementary tables for all figures (see Tables S1-S5), which we refer to in the legends of all main text figures.

L233 Table S2?

Thanks for pointing out this mistake. We have now corrected and checked all references to supplementary tables.

L238 You control for sampling effort for abundance. If possible do the same for richness and Simpson diversity (see earlier comment).

We now include an additional analysis in the supplementary information fitting estimated richness (using the Chao estimator) as a function of land-use intensity and land-use type (Figure S6), showing a similar pattern of response. Since our predictions for Chao-estimated richness are similar, we opted to keep our original species richness metric in the main text, with a reference to this supplementary figure in the Methods (L224, L235) and Discussion (L598) of the main text.

L242 It is only important that the residuals are normally distributed.

We now describe that we log transformed total abundance and Simpson diversity to normalise the residuals (L243).

L242 Log-transformation is suboptimal (O'Hara et al. 2010). Please use a GLMM with proper error distribution (e.g. negative binomial or gamma). For abundance you can directly control for sample effort in such a binomial model.

Thanks to reviewer 2 for pointing potential improvements to our modelling structure. We now include an additional analysis in the supplementary information with total abundance fit using a zero-inflated negative binomial model (Figure S2), which showed a quantitatively similar pattern of response. We introduce this additional analysis in the Methods (L257). We opted to keep our original figure in the main text with a reference to this supplementary analysis in the main text (L258, L374). We did this for two key reasons: 1) using a negative binomial distribution for non-integer data is problematic; and 2) the results are quantitatively similar between our original model (Figure 2) and the negative binomial model (Figure S2).

L243 About the zeros, see my earlier question and if many, you might consider a zero-inflation model.

We now fit a zero-inflated model in the supplementary information for total abundance (Figure S2). Please also see our response to your comment above.

L249 Your random effect controls for the nested structure but not necessarily for the spatial structure. Therefore, you should check for spatial autocorrelation and control for it if necessary.

We now test for study-level spatial autocorrelation in the residuals of our main LUI models (Figure 2) using Moran's I (Figure S4), showing that there is significant spatial autocorrelation in only 2.33% of species richness studies and 4.65% of total abundance studies, less than the 5% that would be expected by chance using a threshold P value of 0.05. We mention this in the main text Results (L379) and in the legend of Figure S4.

L252 All models? This is not relevant for linear models.

We now state that we checked for overdispersion in all species richness models (i.e. all generalised linear models with Poisson errors (L250).

L260 This justification is rather weak. If this is the case, why did you do it (Fig. 2)? I acknowledge that conducting detailed analyses per taxonomic groups for all land-use types would heavily increase the results and the space for discussion needed. However, we already know quite a lot about cropland, but much less on other land-use types. It would be really interesting to expand this analysis at least to pastures and urban areas (or any other subset for which you have a good reason).

Reviewer 2 raises a good point regarding our justification. Thanks for raising this. We have now removed this justification from the methods, as well as our comment on fertiliser in relation to urban environments (following the suggestion of reviewer 3). Given the quantity of additional supplementary analysis included in our revised version, we opted not to explore additional breakdowns for urban and pasture.

L272 Table S3?

Thanks for pointing out this mistake. We have corrected and checked all references to supplementary tables.

L281 Table S20. Many supplementary table are not referred to at all.

Thanks for pointing out this mistake. We now cite all supplementary figures and tables in the main text.

L298 Which statistical software did you use?

Thanks for pointing out this missing citation. We now cite R v.4.0.3 (R Core Team 2020) (L325).

L299 Please provide measures of goodness of fit for your models in the results.

We now include pseudo R squared values for all the models we fit in our paper (Table S18). We also reference this table in the main text (L253).

L336 Please differentiate better for richness and abundance.

We have now restructured the Results section to be better differentiate change in pollinator biodiversity in relation to land-use intensity (L350-379).

L337 and 339 Please check the percentages carefully. It seems that you are drastically overestimating the reduction. It seems that you calculated the percentage as the respective range in Fig. 2. E.g. for abundance change in urban sites: roughly 54% (MU) + 14% = 68%. If this is the case, the difference is reported relative to the baseline instead of the highest value (or lowest intensity) of the respective land use type. E.g. baseline= 10, Urban MU = 15.4 (+54% to baseline), Urban IU = 8.6 (-14% to baseline), span in percent is 68% as reported for abundance, but Urban IU is only a 44% reduction relative to Urban MU (difference = 15.4 - 8.6 = 6.8, which mean a 44% reduction from 15.4).

Thanks to reviewer 2 for pointing out this mistake. For change across the gradient of intensity within a land-use type we now calculate percentage change as suggested (L362-369).

L372 Without error bars for primary vegetation, diversity is higher in minimal use. Please also refer to the respective supplementary tables here and for all subsequent results.

We have reworded this sentence to emphasise that we mean significant difference among cropland intensity classes (L404). We have also included tables in the supplementary information for the model outputs of this model and those of all other models (Tables S6 and S7; and Tables S8-S15).

L375 To avoid the impression that you had significant effects please change to: "weak and insignificant". Since there was no effect, there is no need to show this figure.

We have changed the above to “insignificant for all of species richness, total abundance, and Simpson diversity (Table S17), meaning it was excluded from further analysis” (L416).

L377 Here you are mixing comparisons to baseline and among increasing land-use intensity. Please be more precise. Compared to baseline, you found negative effects in most of the cases, but there is also a positive effect (Hymenoptera, richness). For increasing land-use intensity you found a negative effect only for Hymenoptera for abundance, and a negative effect for Lepidoptera and Passeriformes and a positive for Diptera for richness.

Thanks to reviewer 2 for emphasising the distinction between change within a land-use type and change relative to the baseline. We now more carefully distinguish between change relative to the baseline (L356-361) and change within a land-use type across the gradient of intensity (L362-371). We repeat this structure in the abstract (L14-19) and in the concluding paragraph (L627-629).

L380 All three biodiversity metrics? You said that there was no effect for Simpson diversity.

After subsetting our pollinator data following our expert consultation, all of species richness, total abundance, and Simpson diversity were significant for the Lepidoptera. We state this explicitly in the Results (L422-423).

L384 Please stick to the error bars, there was no effect.

We have now clarified that we mean a significant reduction for only high intensity cropland for the Passeriformes (L428-430).

L387 Why do the biodiversity metrics for the same group not cover the same range in fertilization in Fig. 6?

Biodiversity metrics for each taxonomic group cover 95% of the range of fertiliser values sampled. We reasoned that it would not be appropriate to predict beyond the range for which we have data. We now clarify this point in the figure legend of Figure 5 (previously Figure 6).

L387 This was not the case. The relationship was positive for most taxa and only negative for Hymenoptera and Lepidoptera.

We have amended this first sentence to state that the response to fertiliser application rate was mixed (L431-432).

L391 ff Please provide % changes as we did before and not x-fold changes. Also, make the fertiliser ranges comparable, i.e. refer to changes per 1000 kg/ha for instance.

We now describe changes for fertiliser as percentages for changes per 1000 kg/ha (L434).

L396 Results are missing for the fact that Coleoptera, Apodiformes and Passeriformes increased in most metrics.

We now include specific results for the Apodiformes and Passeriformes (L439-441). Coleopteran response to fertiliser application rate was insignificant for all biodiversity metrics, which we now highlight in the results (L437-439, also see significance of interactions indicated in Figure 5).

L455 Simpson was not significant in most of the cases (only for fertilizer in croplands).

Given that Simpson diversity is significant in our models for difference among taxonomic orders, we have opted to keep this mention of Simpson diversity at the beginning of the Discussion (L477).

L461 The response in the tropics was not much greater than in temperate regions (suggesting that there was an effect in the temperate zone), but it was restricted to the tropics.

We have amended this sentence to state that we “highlight a strongly negative response to land-use intensity in tropical cropland, and no apparent effect of land-use intensity in non-tropical cropland” (L490). We have also amended this statement in the abstract to be consistent (L20).

L463 Do you mean wild plant pollination? There are other areas which are pollination-dependent for crops.

Thanks for clarifying that this is unclear. We have clarified here that we mean “the dominance of animal pollinated plants in tropical environments” (L491), which is consistent with the findings of Ollerton et al (2011) and Rech et al (2016).

L471 You may refer to some homogenization literature.

We have removed this sentence from the end of this paragraph since it largely duplicated content of the previous sentence (L500).

L472 You did not find an overall negative response to land-use intensity (see earlier comments). You need to be much more differentiated, and especially clear when you refer to comparisons with the baseline and when to differences between different levels of intensity independent of the baseline.

We would agree that our original statement here was not consistent with the results of our study. In our revised version we now introduce the Discussion by stating that “land-use intensity is associated with significant changes (both positive and negative) in pollinator species richness, total abundance, and Simpson diversity, particularly for invertebrate pollinators” (L475-477). We have also removed the statement previously at L472, instead specifically mentioning the groups in which we saw declines (L519-521).

L476 Diversity was not maintained it was considerably increased.

We have amended this sentence to state that urban biodiversity was higher at minimal intensity than the primary vegetation baseline (LXXX).

L478 This is a big jump and some restructuring might help.

We agree that the link within this paragraph was not clear. We have now moved our comments on urban sites within the first paragraph (LXXX), and focus on tropical/non-tropical differences alone in this separate paragraph (L484-485).

L480 It would be worth to mention that during the long history of land-use in temperate zones most anthropogenic activities led to an initial increase in biodiversity by generating new habitat types and increasing landscape heterogeneity. Such positive effects are also visible in your results, e.g. when minimal or light use lead to an increase in diversity. You should not only focus on the negative effects but also discuss the positive effects.

We agree that our initial submission could have more clearly acknowledged the nuances of pollinator biodiversity change. We now emphasise increases in pollinator biodiversity in the first paragraph of the Discussion (L480-488). We also stress difference in response in the abstract (L14-15) and the concluding paragraph of the Discussion (L625-630).

L481 Delete “REF”

We have deleted ‘REF’ from this citation (L505).

L486 Do you have a reference for the shorter history?

We have added a reference for the shorter history of intensive agriculture in the tropics (L511).

L489 2002 instead of 202

We have corrected this citation to ‘2020’ (L514).

L491 Do not mix support for temporal declines with your space-for-time results. “we found strong declines [...] on high intensity cropland” and referring first to temporal declines generates the impression that you confirm temporal declines, which is not the case.

We agree that our original opening to this paragraph was misleading. We have now amended this sentence to emphasise again that our study is a space-for-time analysis (L475), and that we found significantly lower abundance on high intensity cropland – i.e. not declines over time (L519-521).

L494 Not clear what you are referring to, baseline or increasing intensity? Actually, you found declines only for Hymenoptera and an increase for Diptera. Compared to baseline, Diptera declined the same as the others.

We have amended this statement to specifically state that “we found significantly lower abundance on high-intensity cropland—relative to primary vegetation—for all insect orders, especially for the hymenopterans, lepidopterans, and coleopterans. We also found significant reductions in insect abundance and richness in response to increasing fertiliser application rate on croplands, particularly for the hymenopterans and lepidopterans” (L519-523). In the subsequent separate paragraph we address increases in biodiversity for the Diptera (L538).

L505 But why do they respond positively with fertilization? Could this be related to larval diet?

Given the insignificant effect for beetles in response to total fertiliser application rate we have amended this sentence. We now state that the positive effects of fertiliser may be buffering the more negative effects of associated intensity. We state that this may be related to larval preference for fertile soils (L534-535).

L508 Please be more precise. There was no effect for richness (instead of ‘less severe’) and even a positive for minimal use compared to the baseline.

At the suggestion of the pollination ecologists we consulted, we have now removed all ant species from the analysis we present in the main text, meaning this statement has been removed.

L515 Two out of four is not 'most'. In addition to Diptera, Coleoptera increased as well (richness and Simpson, likely no response in abundance).

We have removed this sentence from our revised version, starting this paragraph instead with "Dipteran abundance and species richness increased with fertiliser application rate" (L538). Coleopteran response to total fertiliser application rate was insignificant for all of species richness, total abundance, and Simpson diversity (L437-439).

Fig. 1 Please provide information about land-use intensity by colouring the dots e.g. according to LUI or if too messy only to land-use intensity.

We have now coloured the points for each site according the land-use type (Figure 1). We have also now removed all sites of unknown use intensity, which we realised were included here in the original version of this figure.

Fig. 2, 3 and 5 Please provide percentiles also for the baseline.

To calculate our percentiles we calculated the percentage change from the baseline to each factor, for each sample from the covariance-variance matrix. Our overall predictions and percentile confidence intervals therefore account for variation in the baseline, but because all predictions are rescaled to 100% at the baseline, no uncertainty is displayed on the figure. Similar approaches have been used previously in Gibb et al (2020), De Palma et al (2015), and Williams et al (2020). In the supplementary information we also include an additional figure showing how our predictions on cropland relate to a high and low forest cover baseline (Figure S12).

De Palma, A., Kuhlmann, M., Roberts, S.P., Potts, S.G., Börger, L., Hudson, L.N., Lysenko, I., Newbold, T. and Purvis, A., 2015. Ecological traits affect the sensitivity of bees to land-use pressures in European agricultural landscapes. *Journal of Applied Ecology*, 52(6), pp.1567-1577.

Gibb, R., Redding, D.W., Chin, K.Q., Donnelly, C.A., Blackburn, T.M., Newbold, T. and Jones, K.E., 2020. Zoonotic host diversity increases in human-dominated ecosystems. *Nature*, 584(7821), pp.398-402.

Williams, J.J., Bates, A.E. and Newbold, T., 2020. Human-dominated land uses favour species affiliated with more extreme climates, especially in the tropics. *Ecography*, 43(3), pp.391-405.

Table S7 P-values are missing

For all significant models we now include all the summary statistics and associated p values in the supplementary information (Table S6-S15).

O'Hara RB, Kotze DJ (2010) Do not log-transform count data. *Methods in Ecology and Evolution* 1: 118-122.

We now present an additional set of analyses in the supplementary information, in which we fit Simpson diversity and total abundance with a negative binomial model in the R package glmmTMB. For total abundance we fit the model with zero inflation, and for Simpson diversity we fit the model without zero inflation. These models return very similar prediction intervals to the main text (Figure S2), indicating that our original main text approach is robust. We refer to these additional models where relevant in the Methods and Results (L257; L373).

Reviewer 3

In their study “Global effects of land-use intensity on local pollinator biodiversity”, the authors gathered data from 340 studies covering >5000 species judged to be pollinators to model the effects of land-use type and land-use intensification on pollinator species richness, abundance and diversity. Using this very impressive dataset, the authors find that while land-use type only weakly affected pollinator biodiversity, increasing land-use intensity negatively affected pollinator richness and abundance across all land-use types. Moreover, declines in intensified cropland were found to be greater in tropical than in temperate regions, with specialized pollinator clades such as bees and butterflies being particularly negatively affected.

While the individual findings are not totally surprising or outstandingly novel, this study definitely is one of the (if not the) most comprehensive analysis and attempts at drawing general conclusions on the effects of human land-use on pollinator biodiversity at a global scale. Thus, this manuscript may represent a very important contribution to the literature that should attract a great number of readers and interest from ecologists, conservationists and members of the general public interested in pollinator conservation – a very important topic given the great reliance of wild plants and global crop production on animal mediated pollination.

Thanks very much to reviewer 3 for your positive comments on our manuscript. We address each of your comments below, including additional analyses or main text edits where appropriate.

My main concern with this study is the authors’ usage of a baseline to which to compare pollinator responses in richness/abundance/diversity with changing land-use type and land-use intensity (see Figures 2+3). In the respective models, the authors used pollinator data from study sites classified as “minimally used primary vegetation” as a baseline. The results are then presented as %-differences to this baseline (e.g., 20% lower or higher species richness in a given land-use type as compared to primary vegetation). At first glance, this appears to be a very intuitive way to present the data, and to compare pollinator responses across different biodiversity metrics (richness, abundance, Simpson diversity) and across land-use types of varying intensity levels. However, I wondered if primary vegetation is in all cases the ideal baseline for such a %-level comparison. What is striking is, for example, that, on average (i.e. across intensity levels) many land-use types have much higher richness and/or abundance than sites covered with primary vegetation (see e.g. Figure 2). Negative effects of human land-use mainly become apparent at higher levels of land-use intensity. Figure 3 suggests that such positive effects were mainly driven by datasets from the temperate region, where (minimal to light) land-use often supports higher pollinator biodiversity than primary vegetation. It remains unclear how primary vegetation was defined in these temperate studies, but I suspect these are mainly studies conducted in forested ecosystems (e.g., pollinator studies conducted in central and northern Europe). However, given that much of Europe is dominated by cultural landscapes characterized by open ecosystems, I wonder whether choosing such forested ecosystems as the reference for the pollinator biodiversity baseline is really the best choice. For example, the vast majority of endangered bees in central Europe are adapted to open habitats (that are of anthropogenic origin and have been cultivated as such often for centuries), such as dry grasslands, orchards or areas with high amount of bare ground for nesting (often on agricultural land). By comparison, pollinator biodiversity is usually low in forested ecosystems (which is sometimes even the case in the tropics, where flower visitor activity is often much higher at forest edges and in forest clearings because of a lack of flowering undergrowth vegetation in closed rainforests). Hence, while I understand the appeal of having a baseline measurement for pollinator biodiversity, I would like to

see more justification for the choice of (and the habitat types hidden behind) primary vegetation, especially in the context of temperate ecosystems with a long history of human land-use influence.

Reviewer 3 raises an important and interesting point on baselines. Primarily, we would emphasise that our choice of baseline factor (primary vegetation, minimal use intensity) for Figure 2 will not influence the relative difference among predictions. The baseline for Figure 2 could be set for any factor, and the predictions would not move relative to each other. Our choice of baseline factor is therefore made more on precedent, as opposed to an a priori assumption that primary vegetation will always have relatively higher pollinator biodiversity.

Having said the above, reviewer 3 is right to query differences in primary vegetation structure between the tropical and non-tropical regions, given that this may be confounding our predictions. We now use data for global forest cover (Hansen) and terrestrial ecoregions to investigate tropical/non-tropical differences in our minimal use primary vegetation baseline. Specifically, we include Figure S10, which explores the distribution of baseline sites among terrestrial ecoregions, and Figure S11, which maps forest cover for all baseline sites. We also explored how our inferred pollinator responses differ between areas of high and low forest cover (Figure S12), showing that although low forest cover baseline sites do indeed support higher pollinator biodiversity, the relative difference among other land-use types (i.e. cropland) remains largely unchanged. We introduce these analyses in the Methods (254-278), and refer back to them later in the Discussion (L588-595).

In addition, as most of the results are currently expressed as %-differences (to a given baseline). This obscures a bit the underlying absolute differences, which would be nice to see. For example, in Figure 4, model predictions for absolute measures of pollinator richness, abundance and diversity are presented. I was surprised to see that these predictions are, however, on a relatively low level (e.g., averages of species richness of pollinator in temperate cropland ranging between ca. 3.5 and 2.5 species, and in the temperate zone between 3.5 and 4 species; Fig. 4A). This made me wonder how large the absolute differences in species richness and abundance actually are (e.g., in Figs. 2+3+5), when not expressed on a %-differences scale.

Thanks to reviewer for emphasising absolute values of biodiversity in our paper. Absolute values of biodiversity in the PREDICTS database to some extent reflect properties of sampling, as opposed to a true representation of pollinator biodiversity within a given ecosystem. Hence, we chose to focus on % changes in sampled biodiversity to show responses of pollinator biodiversity to land use and land-use intensity. Previous studies have taken this same approach of examining biodiversity differences as a percentage change as opposed to absolute values:

De Palma, A., Kuhlmann, M., Roberts, S.P., Potts, S.G., Börger, L., Hudson, L.N., Lysenko, I., Newbold, T. and Purvis, A., 2015. Ecological traits affect the sensitivity of bees to land-use pressures in European agricultural landscapes. *Journal of Applied Ecology*, 52(6), pp.1567-1577.

Gibb, R., Redding, D.W., Chin, K.Q., Donnelly, C.A., Blackburn, T.M., Newbold, T. and Jones, K.E., 2020. Zoonotic host diversity increases in human-dominated ecosystems. *Nature*, 584(7821), pp.398-402.

Other comments:

L17: In the discussion flies are presented as a pollinator group that is relatively robust to land-use, so I was surprised to see them mentioned here together with the more sensitive bees, butterflies, etc.

Thanks for pointing out this mistake. At the suggestion of reviewers 1 and 2, we now emphasise increases in some pollinating groups, since this better represents the nuances of pollinator biodiversity change.

L29: What do you mean with “mainstream media”? I wonder about the choice of wording (but I am not a native speaker – am just curious because of Trump’s usage of such words).

Thanks to reviewer 3 for pointing out our poor choice of language here. We have changed this term from “mainstream media” to “media” (L30).

L42: Perhaps include “US” in addition to “dollars” to clarify.

We have added “US dollars” here for clarity (L45-46).

L47: Not all of the 75% of globally important food crops do fully rely on animal pollination (some show only very modest yield increase), but they do so at least partially (and some fully). I suggest including “at least partially” before “rely” to make this clear.

We have added “at least partially” to make this point clear (L50).

L89-96: In addition, pollinator increases in open habitats may simply be driven by greater availability of floral resources and nesting sites (for ground-nesting pollinators, e.g., 66%-75% of solitary bees in central Europe).

We now mention floral availability in the context of urban and open habitat benefits (L96-97), and nesting behaviour in the context of species traits (L100).

L101-103: In addition, there remain uncertainties associated with taxonomic and geographic biases that further hinder general conclusions on pollinator responses to land-use change globally (de Palma et al. 2016, Scientific Reports)

We now cite De Palma et al (2016) in the Introduction, in the context of uncertainty around models based on restricted geographical areas (L39).

L112: Please also include the numbers of studies included from these geographic regions. Looking at Figure 1, you did a remarkable job to also include as many studies from Africa, Asia and South America as possible. Nonetheless, there still appears to be a geographic bias in favor of the global North (as is usually unavoidable in such global datasets).

We now include the percentage of sites in each of these geographic regions (L343-345). In this revised version we now also include an additional jack-knife analysis, showing that single continents do not markedly influence our predictions. We describe this jack-knife analysis in the Methods (L263-265) and Discussion (L603), with an additional figure in the supplementary information (Figure S3).

L115: I agree with the hypothesis that pollinator biodiversity may respond more negatively to land-use (intensification) in the tropics, but the point was not very well developed in the introduction. You briefly mention the potential greater susceptibility in the tropics (L98-101). In addition, you may also wish to include arguments from your discussion, e.g., the point that temperate ecosystems have been shaped by humans for a much longer time and in a more persistent fashion than most tropical regions. Think of cultural landscapes (small fields, orchards, meadows, hedgerows) that harbor much

greater biodiversity than modern agricultural landscapes (simplified and intensively used cropland) or natural ecosystems (forests), and which in fact even favored a great number of synanthropic species. By contrast, tropical biodiversity is often not adapted to human land-use.

Thanks for pointing out that this background for tropical/non-tropical differences is unclear. In our revised version we now include an expanded paragraph exploring the potential reasons behind difference in response between geographical zones (L104-119), which we explore in more detail in the Discussion (L501-516).

L237: Was this index of sampling effort used in any of the following models? I could find mentioning of random effects, but not of this variable.

We now mention in the methods that we worked only from the sampling effort adjusted measurements (L230-231).

L245: It surprised me that most recorded measurements of pollinator abundance were not integer values – where these then expressed as densities, or why was this?

We now describe in the methods that many measurements were recorded as densities, which in conjunction with our sampling effort correction means many measurements are analysed as non-integer values (L231-233).

L254-255 + L273-275 + L294-296: I could not make sense of these sentences “We tested each model against an intercept model, and excluded any biodiversity metric for which ΔAIC was greater than ~ 2 ”. With “biodiversity metric”, do you mean pollinator richness/abundance/diversity? And how do you exclude these? (you can exclude predictors from your model but not the response variable – i.e., the biodiversity metric?). I believe you mean you discarded the model in case the null model (including only the intercept) performed better (based on the AIC score)?

Thanks to reviewer 3 for pointing that this is unclear. We now state that we “discarded any main model for which AIC was greater than the null model” (L253, L297). We have also updated our table of AIC values (Table S17) to reflect this.

L261: I do not agree that fertilizer application is less important in urban environments. Particularly in the tropics, urban farming plays a big role for food security – and here, of course, also fertilizers are applied, if available. In general, I was surprised that not all land-use intensity levels were compared across all land-use types – although lack of data may certainly be an explanation.

We agree that our justification here regarding fertiliser application in urban environments was weak. We have now removed this statement from our revised version. As you suggest our reason for combining land-use intensity with land-use type as a single variable was to account for low site representation. However, we recognise that a model fit separately is useful for teasing out the relative effects of intensity and type. In the methods we now describe an additional analysis in which we fit type and intensity separately (L236-238). In the Results we refer to these models (L350-356), and in the supplementary information we include ANOVA tables for each model (Table S20).

L265-266: Please see my thoughts concerning this choice of baseline above.

Please see our previous response to your comment on baselines. We now also include a reference here to our 3 figures in the supplementary information (Figures S10, S11, and S12) which provide more information on this baseline (L273-278).

L276: I very much liked the additional focus on fertilizer rate effects in cropland. This made me wonder whether other such data are available, e.g., on pesticide application (at a global scale)? This would be particularly interesting, as these additional analyses provide a much more direct link of land-use intensification to pollinator biodiversity – which is a bit harder to grasp when only presented with the land-use intensity classes.

Thanks for your positive comments on our fertiliser application analysis. We now include an additional analysis in the supplementary information investigating response to pesticide application rate, using PEST-CHEMGRIDS (Maggi 2019) (Figures S8 and S9). We introduce this analysis in the Methods (L321-324) and then refer back to it in the Discussion (L554).

L303ff: Honestly, I found the whole system of classifying the pollinators into confidence levels rather confusing. This was certainly necessary and important when gathering the study dataset. But the classification is hard to grasp – because higher numbers do not necessarily indicate higher certainties (the opposite appears to be rather the case), and because there appears to be a break in the confidence changes between 1-4 and 5.1-5.4 (so no real logical progression from 4 to 5.1). Lastly, I wondered if you also tried some sensitivity tests of your model outcomes based on different levels of pollinator confidence – e.g., only using data with pollinator classifications based on high levels of confidence?

We agree that our system of classifying confidence was unclear in our initial submission. We have now removed the entirety of the extrapolated confidence classification system from our revised version. Instead, we now include only our system of classification for direct level confidence, and any extrapolated taxa we simply define as likely pollinators (L177). As an alternative check against uncertainty, we consulted the expert opinion of 7 pollination ecologists (please see response to reviewer 1), and removed or added any taxa at their suggestion.

L311-312: Please also provide numbers or %-values of sites across these geographic regions, so that differences in geographic coverage become clearer.

We now include the percentage of sites in each of these geographic regions (L343-345).

L332: Figure 1 - Reptiles are impossible to see in panel C – perhaps use a log-scale instead.

Thanks for stating that the number of reptiles is unclear. We now state in the legend for Figure 1 that the reptiles are represented by only 5 species with direct confidence, with a reference to Table S1 in which we show the number of pollinating species in this figure. We decided not to use a log-scale here given that this may be difficult to interpret as a stacked bar plot.

L425: Figure 2 – Percentiles are much greater for predictions in the temperate sites. Is this only due to larger variation because of the greater number of study sites (as compared to number of tropical study sites), or also a result of more divergent pollinator responses in temperate regions?

We now include an additional figure in the supplementary information indicating that the greater variation in temperate (now referred to as non-tropical) sites is not strongly predicted by sample size (Figure S14). We sampled 1000 sites from each of the tropical and non-tropical sites a total of 100 times, and then for each sample of 2000 (tropical and non-tropical) fitted total abundance as a function of land-use intensity, geographical zone, and their interaction. We then plotted the distribution of the size of the 95% confidence intervals for all models, indicating that responses in

the non-tropical zone are more variable independent of sample size. We describe this additional analysis in the methods (L279-284) and refer back to it in the Discussion (L605-607).

L440: Figure 4 – Are all of these model predictions statistically significant? If not, please use dashed lines to indicate non-significant relationships.

All of the relationships here were non-significant. At the suggestion of reviewer 2 we have now removed this figure from the revised version.

L452: Figure 6 – Again, I wondered about significant vs. non-significant effects.

We have now added in p values for the interaction terms of Figure 5.

L461: Word too much/missing in this sentence.

We have now reworded this sentence at the suggestion of reviewer 2.

L462: Greater reliance of what? Angiosperm plants in general, crops, or both?

We have rephrased this sentence at the suggestion of reviewer 2 to more accurately reflect the analysis of Rech et al (i.e. “This is an important result, given the dominance of animal pollinated plants in tropical environments”) (L490).

L468: Perhaps worthwhile to nonetheless mention studies that highlight the importance of flies and other non-bee pollinators for crop pollination (e.g., Rader et al., PNAS).

We include an additional analysis in the supplementary information in which Figure 3 is subset for the main crop pollinating groups (Figure S13). We introduce this analysis in the Methods (L284-287), and refer back to it in the Discussion of the main text (L574).

L481: Delete “REF”.

We have deleted ‘REF’ from this citation.

L489: Full publication year for Williams et al. missing.

We have corrected this citation to ‘2002’ (L514).

L512: This is an interesting finding, and good that you did this additional analysis. Could you also quantify the “much stronger response for abundance” here in the main text – e.g., by expressing the difference in %?

At the suggestion of the group of pollination ecologists we consulted, on this revised version we have opted to remove ants from all figures of the main text.

L546ff: I appreciate the discussion of study limitations. As outlined before, it would be important to also briefly discuss the choice of the baseline habitat, and how this may affect biodiversity comparisons to anthropogenic land-use types. Even though of anthropogenic origin, many agricultural habitat types with minimal usage are of highest conservation concern (e.g., dry grasslands), and harbor many more rare and common pollinator species (e.g., wild bees) than natural ecosystems with primary vegetation.

Thanks again to reviewer 3 for highlighting the importance of this limitation. We now explicitly mention this limitation in our Discussion (L588-595), and refer to the supplementary Figures S10, S11, and S12, which provide further information on our baseline.

Reviewer #3 (Remarks to the Author):

I have now received the revised manuscript and the authors' response letter - and I am happy to write that they did a great job at addressing all of my previous concerns and comments. Congratulations!

PS: Note that in the Supplementary .pdf-file, Fig S3 and S10 were not fully readable because of the page orientation

Response to reviewers

Reviewer 2

The authors excellently addressed all my points and explained well when they deviated from my suggestions. The discussion is more balanced now and better reflects the results. In this form this is a strong manuscript likely highly interesting for a broad readership. I only have a few, mostly editorial comments.

Thanks very much to reviewer 2 for your positive comments on our revised manuscript. We have now addressed each of your editorial comments in red below.

L193 Change to “kept if they”

We have corrected this as “kept if they” (L564).

L341 Delete space between 12, and 170

We have deleted this space between “12” and “170” (L161).

L368 Delete “for”

We have removed “for” from the end of this line (L204).

L377 Do you mean Table S6 instead of Table 6?

Thanks for pointing out the mistake. This citation should be to Figure S6 (now Supplementary Figure 4)

L408 I don't understand the first part of this sentence. Please reword.

We have amended this sentence to clarify our meaning (L250).

L525 There are also diet generalist butterflies. Please add “many butterflies”

We have amended this statement to “many butterflies” as suggested (L394).

L558 Might it be due to simple correlations between fertilizer and pesticides?

We explicitly state that the similar response for fertiliser and pesticide application rate may be due to correlation (L422).

L583ff Please add to the limitations that your explanatory power of LUI is rather low in almost all models. There are very low marginal pseudo R-squares while the conditional R-squares are quite high, i.e. there is quite some variation explained by the random effects but less by the fixed. You may add that this might have been expected due to the nature of the data in general and since you aggregated ecologically very different taxa.

We have now added an additional statement in the limitation of the discussion highlighting that the explanatory power of our models is low (L456). We emphasise that the aim of our analysis was not to predict pollinator biodiversity for a specific location, but rather to investigate general trends in the direction and magnitude of change.

Supplementary figures and tables should appear in consecutive order.

We have changed the order of the Supplementary Figures and Tables to appear consecutively.

Reviewer 3

I have now received the revised manuscript and the authors' response letter - and I am happy to write that they did a great job at addressing all of my previous concerns and comments. Congratulations!

Thanks very much to reviewer 3 for your positive comments on our manuscript.

PS: Note that in the Supplementary .pdf-file, Fig S3 and S10 were not fully readable because of the page orientation

We have changed the sizing and orientation of the previous Supplementary Figures 3 and 10 (now 2 and 12).